# Ait1 regulates TORC1 signaling and localization in budding yeast

**Ryan L Wallace†, Eric Lu†, Xiangxia Luo, Andrew P Capaldi***

Department of Molecular and Cellular Biology, University of Arizona, Tucson, United States

**Abstract** The target of rapamycin complex I (TORC1) regulates cell growth and metabolism in eukaryotes. Previous studies have shown that nitrogen and amino acid signals activate TORC1 via the highly conserved small GTPases, Gtr1/2 (RagA/C in humans), and the GTPase activating complex SEAC/GATOR. However, it remains unclear if, and how, other proteins/pathways regulate TORC1 in simple eukaryotes like yeast. Here, we report that the previously unstudied GPCR-like protein, Ait1, binds to TORC1-Gtr1/2 in *Saccharomyces cerevisiae* and holds TORC1 around the vacuole during log-phase growth. Then, during amino acid starvation, Ait1 inhibits TORC1 via Gtr1/2 using a loop that resembles the RagA/C-binding domain in the human protein SLC38A9. Importantly, Ait1 is only found in the *Saccharomycetaceae/codaceae*, two closely related families of yeast that have lost the ancient TORC1 regulators Rheb and TSC1/2. Thus, the TORC1 circuit found in the *Saccharomycetaceae/codaceae*, and likely other simple eukaryotes, has undergone significant rewiring during evolution.

## Editor's evaluation

Genetic analysis of the yeast *S. cerevisiae* identifies a role for the vacuolar GPCR Ait1 in the regulation of TORC1. The mechanism of Ait1 function is mediated by regulation of the small GTPases Gtr1/2. This finding identifies novel signaling modality in yeast for the control of TORC1 function.

**\*For correspondence:**
capaldi@email.arizona.edu

†These authors contributed equally to this work

**Competing interest:** The authors declare that no competing interests exist.

## Introduction

To function correctly, cells have to set their growth rate based on a wide array of nutrient, stress, and hormone signals. In eukaryotes, this fine-tuned control depends—in a large part—on a single, highly conserved, signaling hub called the target of rapamycin kinase complex I (TORC1) (*González and Hall, 2017*; *Loewith and Hall, 2011*; *Liu and Sabatini, 2020*). In the presence of pro-growth hormones and abundant nutrients, TORC1 drives growth by activating protein, ribosome, lipid, and nucleotide synthesis (*González and Hall, 2017*; *Loewith and Hall, 2011*; *Liu and Sabatini, 2020*; *Huber et al., 2009*; *Robitaille et al., 2013*; *Hsu et al., 2011*; *Peterson et al., 2011*; *BenSahra and Manning, 2017*; *BenSahra et al., 2016*; *Kim et al., 2011*; *Kamada et al., 2000*). In contrast, when nutrient or hormone levels drop, TORC1 is repressed, causing cells to switch from anabolic to catabolic metabolism, and eventually enter a quiescent state (*Düvel et al., 2010*; *Barbet et al., 1996*).

TORC1 is regulated by a sophisticated signaling network that, in humans, includes two well-defined channels:

(1) Growth factor and mitogen signals are transmitted to TORC1 through a GTPase activating protein (GAP) called the tuberous sclerosis complex (TSC) (*Manning et al., 2002*; *Inoki et al., 2002*). In the presence of pro-growth hormones (such as insulin), TSC is repressed, triggering accumulation of the active, GTP-bound, form of Rheb (*Menon et al., 2014*; *Dibble and Manning, 2013*). GTP-Rheb then binds to TORC1 on the lysosomal membrane, driving a conformational change that increases

TORC1 activity (*Menon et al., 2014*; *Yang et al., 2017*). The AMP-activated protein kinase (AMPK) also signals to TORC1 via TSC (as well as the TORC1 subunit Kog1/Raptor) to ensure TORC1 is inhibited when ATP levels fall (*Inoki et al., 2003b*; *Inoki et al., 2003a*; *Gwinn et al., 2008*).

(2) Amino acid (and glucose) signals are transmitted to TORC1 via a heterodimeric pair of GTPases, consisting of RagA or B and RagC or D, that are tethered to the lysosomal membrane by the Ragulator complex (*Sancak et al., 2010*; *Sancak et al., 2008*; *Kim et al., 2008*; *BarPeled et al., 2012*; *Efeyan et al., 2013*). In the presence of adequate nutrients, RagA/B and C/D are in their GTP- and GDP-bound forms, respectively, and bind tightly to TORC1 to keep it on the lysosomal membrane and near Rheb (*Menon et al., 2014*; *Sancak et al., 2010*; *Sancak et al., 2008*; *Kim et al., 2008*; *Rogala et al., 2019*). However, when amino acid levels fall, the large multiprotein GAP, GATOR1/2, drives RagA/B to the GDP-bound form, triggering the release of TORC1 from the lysosome so that it cannot be activated by Rheb (*BarPeled et al., 2013*).

GATOR1/2, in turn, is regulated by at least three different amino acid-binding proteins to ensure that cell growth halts during starvation: the leucine sensor Sestrin2 (*Wolfson et al., 2016*; *Saxton et al., 2016b*); the arginine sensor CASTOR1 (*Saxton et al., 2016a*; *Chantranupong et al., 2016*); and the methionine—or more specifically *S*-adensylmethionine (SAM)—sensor SAMTOR (*Gu et al., 2017*). Arginine signals are also transmitted to the Rags via SLC38A9, an amino acid transporter in the lysosomal membrane (*Wang et al., 2015*; *Castellano et al., 2017*).

Outside of humans, however, much less is known about TORC1 regulation. The amino acid sensors discussed above are only fully conserved in vertebrates (*Liu and Sabatini, 2020*; *Tatebe and Shiozaki, 2017*), and while Rheb/TSC and the Rags/GATOR are ancient TORC1 pathway components—likely present in the last common eukaryote—many yeasts, worms, plants, and protists/excavata have lost Rheb and TSC (*Liu and Sabatini, 2020*; *Tatebe and Shiozaki, 2017*). It therefore seems likely that: (1) there are conserved TORC1 pathway components that remain to be discovered, and (2) simple eukaryotes have evolved novel, currently unknown, nutrient sensing and TORC1 control mechanisms to replace Rheb/TSC.

One well-studied organism with a TORC1 signaling network that appears to have diverged significantly from that in humans is the budding yeast, *Saccharomyces cerevisiae*.

*S. cerevisiae* has two GTPases, Gtr1 and Gtr2, that are homologs of RagA/B and RagC/D, respectively (*Binda et al., 2009*; *Dubouloz et al., 2005*). Furthermore, Gtr1/2 are tethered to the vacuole (the yeast equivalent of the lysosome) by a complex that is very similar—but not obviously homologous to—the Ragulator, called Ego1, Ego2, and Ego3 *Powis et al., 2015* (*Zhang et al., 2019*; *Zhang et al., 2012*). The GATOR1/2 GAP that acts upstream of the Rags is also conserved in yeast, and made up of Npr2, Npr3, and Iml1 (the GATOR1 equivalent, known as SEACIT) and Rtc1, Mtc5, Sea4, Seh1, and Sec13 (the GATOR2 equivalent, known as SEACAT) (*Panchaud et al., 2013*; *Neklesa and Davis, 2009*; *Chen et al., 2017*; *Laxman et al., 2014*; *Algret et al., 2014*). However, *S. cerevisiae* do not have SLC38A9, Sestrins, CASTOR, or SAMTOR, and are also missing TSC1/2 and functional Rheb (*González and Hall, 2017*; *Liu and Sabatini, 2020*; *Tatebe and Shiozaki, 2017*).

In line with the expectation that there are differences between TORC1 signaling in yeast and humans, we recently discovered that glucose and nitrogen starvation cause TORC1 in *S. cerevisiae* to move from its position distributed around the vacuolar membrane to a single body on the edge of the vacuole (*Hughes Hallett et al., 2015*; *Sullivan et al., 2019*). Adding the missing nutrient back to the cell—even in the presence of cycloheximide—then reverses the process (*Hughes Hallett et al., 2015*). We also found that TORC1-body formation is initiated by inactivation of Gtr1/2 and requires an interaction between TORC1 and the recently identified TORC1 regulator Pib2 (*Hughes Hallett et al., 2015*; *Sullivan et al., 2019*; *Varlakhanova et al., 2017*; *Kim and Cunningham, 2015*; *Ukai et al., 2018*; *Tanigawa and Maeda, 2017*; *Michel et al., 2017*). TORC1 agglomeration, itself, is then driven by two glutamine-rich, prion-like domains in the TORC1 subunit Kog1/Raptor, and ultimately functions to increase the threshold for TORC1 reactivation (*Hughes Hallett et al., 2015*). In other words, the formation of TORC1 bodies helps to ensure that cells commit to the quiescent state when they have been starving for a significant period of time. Interestingly, the prion-like domains in Kog1/Raptor are found in yeast species and worms that are missing the TSC genes, but are absent from *S. pombe* and higher organisms that do carry the TSC genes (*Hughes Hallett et al., 2015*). This suggests that organisms use either TSC and Rheb, or TORC1-body formation, alongside Gtr1/2 (Rag proteins) to control TORC1 activity (*Hughes Hallett et al., 2015*).

Here, to learn more about TORC1 regulation in yeast, and other simple eukaryotes, we map the TORC1 interactome in *S. cerevisiae* in a wide range of stress and starvation conditions. These experiments lead to the identification of numerous new TORC1 regulators, the most notable of which are a putative phosphate channel, Syg1, and a previously unstudied GPCR-like protein, Ydl180w, that we have named **Ait1** (**A**mino acid-dependent **I**nhibitor of **T**ORC**1**). In follow-up experiments, we show that Ait1 is required to hold TORC1 in its native position around the vacuolar membrane during log-phase growth. We also show that Ait1 is required for TORC1 inhibition during amino acid starvation in cells expressing Gtr1 and/or Gtr2. Interestingly, Ait1 is only found in the *Saccharomycetaceae* and *Saccharomycodaceae*. The yeast species within these related families—which include the pathogen *Candida glabrata*—are unique in that they (1) have highly divergent Rheb, or no Rheb, (2) are missing TSC2 and/or TSC1, and (3) have prion-like domains in the TORC1 subunit Kog1/Raptor (*Tatebe and Shiozaki, 2017*). Thus, an ancestor of the *Saccharomycetaceae/codaceae* gained the novel TORC1 regulator, Ait1, at around the same time it lost functional Rheb and TSC1/2 (approximately 200 million years ago) (*Tatebe and Shiozaki, 2017*; *Shen et al., 2018*), to aid in amino acid signaling and appropriate TORC1 localization. We suggest that similar rewiring of the TORC1 pathway likely occurred during the evolution of many other simple eukaryotes, and that Ait1 represents an important new drug target in yeast.

## Results

### The TORC1 interactome in budding yeast

As a first step toward building a map of the TORC1 regulatory network, we developed an immunopurification protocol that makes it possible to capture and identify TORC1 interactors. Cells carrying Kog1-FLAG, and in parallel cells carrying Kog1-HA, were grown to log-growth phase, or grown to log-growth phase and transferred into stress or starvation medium, rapidly filtered and flash frozen. The cells were then lysed and treated with the short (12 Å) cleavable crosslinker dithiobis(-succinimidyl propionate) (DSP) and the nonionic detergent digitonin (*Murley et al., 2017*). The supernatants from the Kog1-HA and Kog1-FLAG strains were then immunopurified in parallel on anti-FLAG columns, the crosslinkers broken, and the samples analyzed using mass spectrometry.

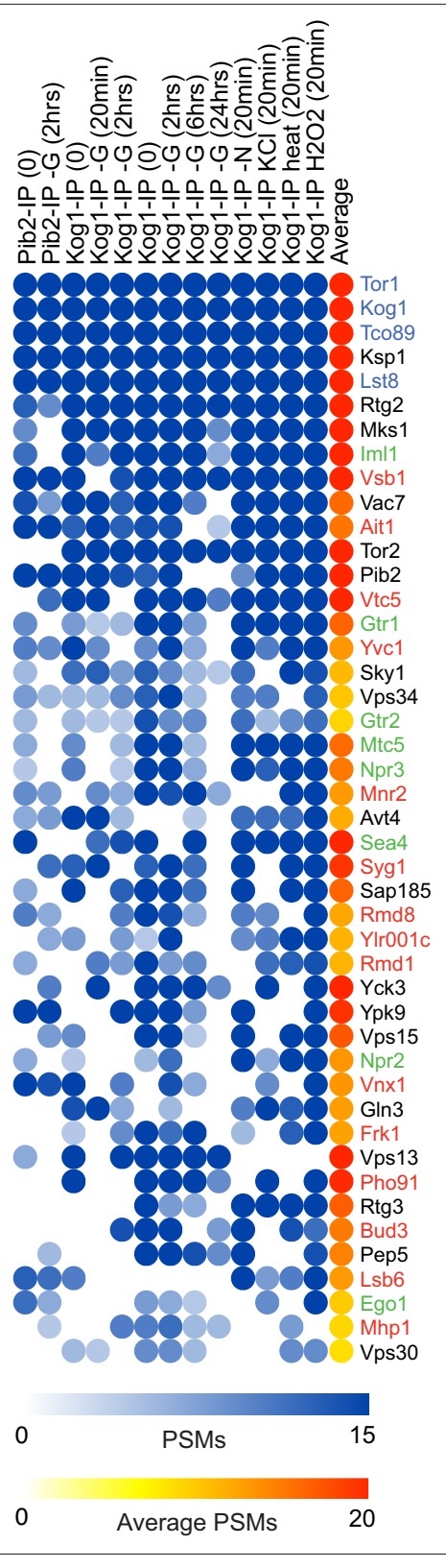

**Figure 1.** The target of rapamycin complex I (TORC1) interactome in budding yeast. Blue circles show the number of background corrected peptide spectral maps (PSMs) from each protein identified in a specific

*Figure 1 continued on next page*

*Figure 1 continued*

Kog1 or Pib2 immunopurification, while the red to yellow scale shows the average number of PSMs across all experiments. The figure shows data for the top 45 TORC1 interactors (those identified in seven or more immunopurifications). The complete dataset is in **Supplementary file 1**.

Proteins with at least twofold higher abundance in the true IP (Kog1-FLAG) versus the mock IP (Kog1-HA), and with at least seven peptide spectral maps in the true IP, were scored as potential interactors. In total, we immunopurified Kog1 in 11 different experiments (in nine conditions) and ran a similar experiment to identify Pib2 interactors in two conditions. These purifications led to the identification of over 200 interactors, 138 of which were identified in four or more experiments (**Supplementary file 1**). At the top of this list are 45 proteins identified in 7 or more experiments, including: (1) all four subunits of TORC1 (blue names, **Figure 1**); (2) Gtr1, Gtr2, and Ego1, all three subunits of SEACIT, and two subunits of SEACAT (green names, **Figure 1**); (3) a variety of proteins that have been shown to play some role in TORC1 signaling previously, including several TORC1 substrates (**Huber et al., 2009**; **Liu et al., 2003**; **Breitkreutz et al., 2010**) (black names, **Figure 1**); and (4) 15 proteins that have, to the best of our knowledge, not been connected to TORC1 signaling previously, or in many cases, studied at all (red names, **Figure 1**). Several of these new interactors—including Ydl180w/Ait1, Vsb1, and Vtc5—form interactions with TORC1 that are as tight, or tighter, than those between TORC1 and Gtr1/2, as judged by the amount of material captured in the purification (**Figure 1**).

To learn more about the interaction between TORC1 and the novel interactor Ait1 (a previously unstudied GPCR-like protein located in the vacuolar membrane **Genome Resources, 2020**; **UniProt, 2019**; **Figure 2**), we also immunopurified GFP-Ait1 after crosslinking, and mapped the interactors, as described above for Kog1. These experiments showed that the TORC1 subunits Kog1, Tor1, and Tco89 are among the most abundant proteins captured in an Ait1 purification, indicating that there is a close interaction between TORC1 and Ait1 (**Figure 3**, **Figure 3—figure supplement 1**).

## Impact of TORC1 interactors on TORC1-body formation

To examine the impact that the new TORC1 interactors have on TORC1 signaling, we measured Kog1-YFP localization during nitrogen starvation in a collection of strains, each missing one of the top 50 proteins identified in our immunopurification experiments (excluding interactors that were examined in our previous studies **Hughes Hallett et al., 2015**; **Sullivan et al., 2019**). These experiments showed that—as expected (**Sullivan et al., 2019**) — many of the known TORC1 interactors are important for TORC1-body formation, including the SEACIT/CAT subunits Iml1, Seh1, Sea2/Rtc1, and Npr3 (**Figure 4**). These experiments also showed that several of the previously unknown TORC1 interactors have a profound impact on TORC1-body formation and/or TORC1 localization. Specifically, deletion of Vnx1 (a vacuolar monovalent cation/proton antiporter; **Wilson et al., 2018**) or Syg1 (a putative phosphate channel in the vacuolar membrane; **Genome Resources, 2020**) blocks TORC1-body formation, just like deletion of Npr2, Npr3, Iml1, or Pib2 (**Figure 4** and **Sullivan et al., 2019**). The most striking result, however, was found in the *ait1Δ* strain: Deletion of Ait1 causes TORC1 to move into a body, even during log-phase growth in nutrient replete medium (**Figure 4** and **Figure 5A, B**).

To learn more about Ait1 function, we next measured the impact that deleting Ait1 has on Kog1-YFP localization in strains carrying mutations that block, or promote, TORC1-body formation (**Hughes Hallett et al., 2015**; **Sullivan et al., 2019**; **Figure 5C**). These experiments revealed that deletion of Ait1 completely overrides the severe defects in TORC1-body formation caused by (1) locking Gtr1 in its active, GTP-bound, conformation (GTR1$^{Q65L}$ or Gtr1$^{on}$ for short), (2) deleting the Gtr1 inhibitor Npr2, or (3) deleting the TORC1-binding protein and regulator Pib2 (**Figure 5C**). However, deletion of Ait1 does not rescue TORC1-body formation in a strain carrying Q to A mutations in the two prion-like domains of Kog1 (*Prm1 + 2*, **Figure 5C**). Thus, Ait1 acts at, or below, the level of Gtr1/2 and Pib2 to hold TORC1 in its native position (distributed around the vacuolar membrane) in nutrient replete conditions. This tethering effect is then lost, or overridden, in starvation conditions.

In contrast to its influence on TORC1, Ait1 does not have a dramatic impact on the localization of the TORC1-binding proteins Gtr1/2 and Pib2, as judged by images of Gtr1-YFP and GFP-Pib2 (**Figure 5—figure supplement 1**). Instead, Pib2 and Gtr1 remain distributed around the vacuolar

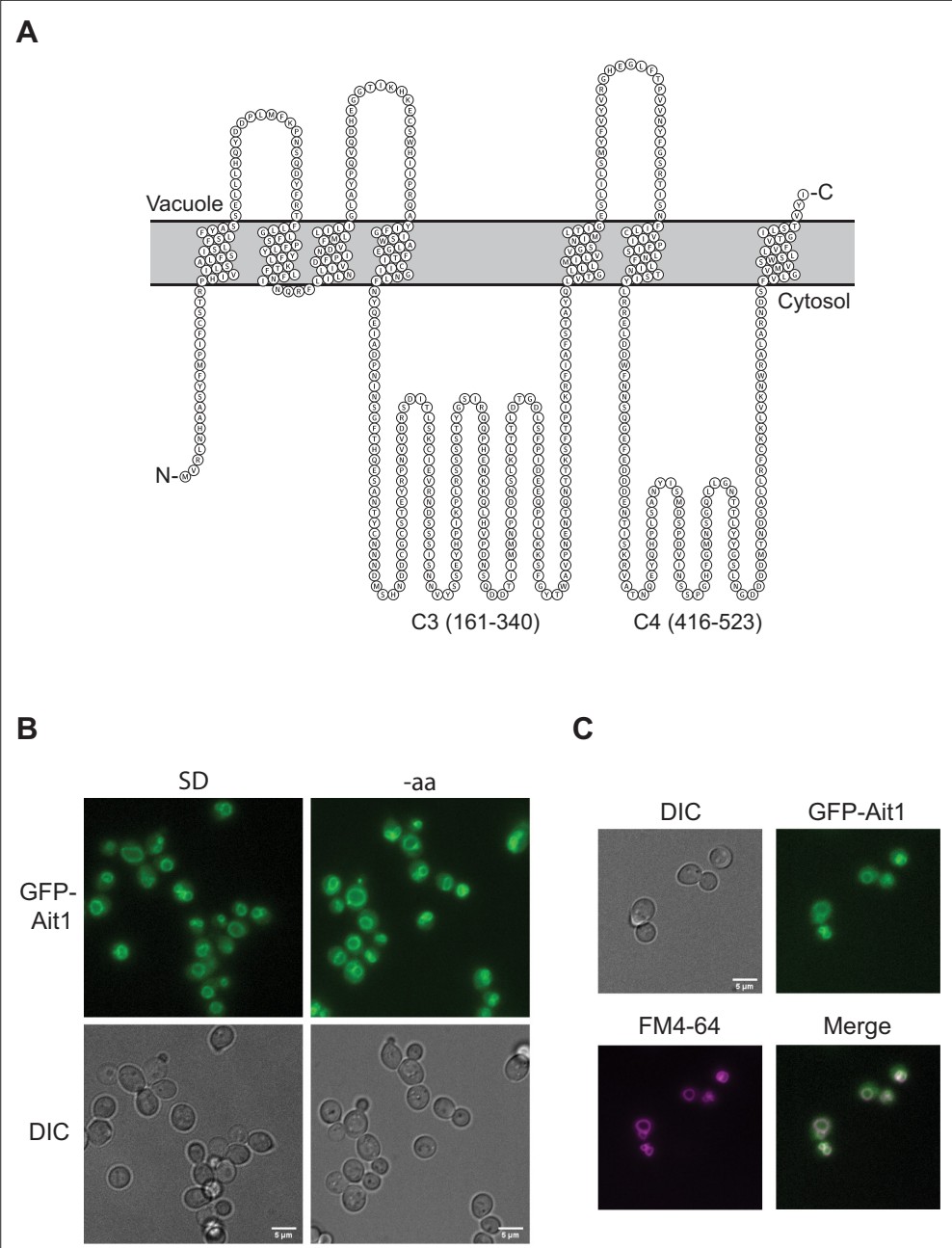

**Figure 2.** Ait1 is a putative seven-helical transmembrane (GPCR-like) protein that localizes to the vacuolar membrane. (**A**) The predicted topology of Ait1/Ydl180w from Protter 1.0 (*Omasits et al., 2014*). The two large cytosolic loops in Ait1, both of which are predicted to be intrinsically disordered (C3 and C4), are labeled. (**B, C**) GFP-Ait1 localizes to the vacuolar membrane, as shown by the overlap between the GFP-Ait1 signal and the vacuolar membrane stain FM4-64 signal and does not relocalize in amino acid starvation (shown), or other starvation conditions (not shown).

membrane in the *ait1Δ* strain, albeit with additional enrichment in foci on the edge of the vacuole (movement that is likely driven by TORC1 agglomeration; *Figure 5—figure supplement 1*).

## Ait1 inhibits TORC1 during amino acid starvation

To test if Ait1 regulates TORC1 signaling, we followed the phosphorylation of a downstream reporter of TORC1 activity, Rps6 (*Chen et al., 2018*; *Yerlikaya et al., 2016*), in wild-type and *ait1Δ* strains. These experiments showed that deletion of Ait1 almost completely blocks TORC1 repression during

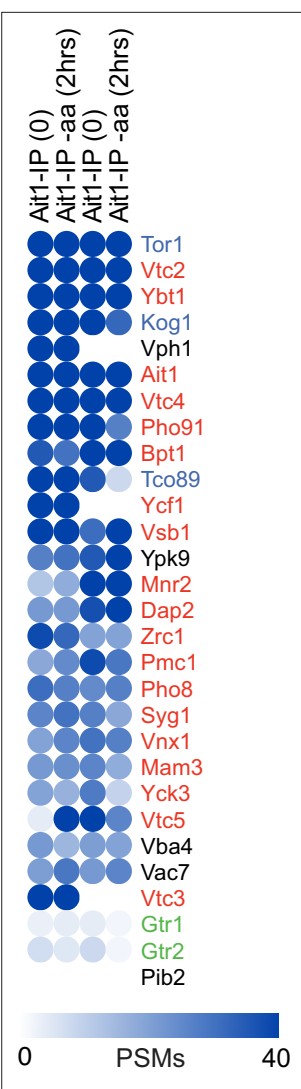

**Figure 3.** The Ait1 interactome. Blue circles show the number of background corrected peptide spectral maps (PSMs) for the top 25 proteins identified in the GFP-Ait1 immunopurification (based on the average number of PSMs in the four experiments), along with the data for Gtr1, Gtr2, and Pib2 for comparison.

The online version of this article includes the following figure supplement(s) for figure 3:

**Figure supplement 1.** Target of rapamycin complex I (TORC1) interacts with Ait1.

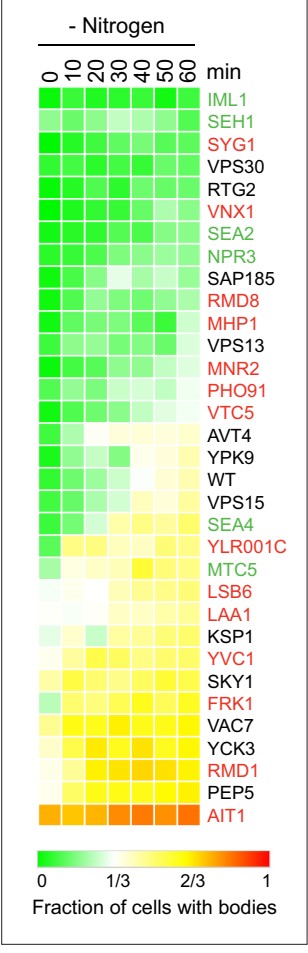

**Figure 4.** Target of rapamycin complex I (TORC1)-body formation during nitrogen starvation in strains missing key TORC1 interactors. Each square on the heat map shows the fraction of cells with a Kog1-YFP focus/body at a specific timepoint, calculated by examining the images of >200 cells, per strain, per timepoint. Replicate experiments confirmed the severe defects in the *syg1Δ*, *vps30Δ*, *rtg2Δ*, and *vnx1Δ* strains (<15% bodies after 1 hr of nitrogen starvation). These follow-up experiments also revealed dramatic variation in the results for *vsb1Δ*cells (even comparing between colonies) leading us to drop the strain from our analysis.

amino acid starvation (in a standard lab strain, *Figure 6*); a phenotype similar to that seen in a strain with Gtr1 locked in its active, GTP-bound, state (*Figure 6*). In contrast, Ait1 does not impact TORC1 inhibition during complete nitrogen starvation (*Figure 6—figure supplement 1*).

Previous studies have shown that leucine is the primary amino acid activating TORC1 via Gtr1/2 (*Bonfils et al., 2012*). We therefore tested if Ait1 is also required for TORC1 inhibition in cells starved for leucine. This was the case; an *ait1Δ* strain has over 80% TORC1 activity after 6 hr of leucine starvation, as judged by Rps6 phosphorylation (*Figure 6*), and nearly 100% TORC1 activity as judged by the phosphorylation of the direct TORC1 substrate (*Figure 6—figure supplement 1*).

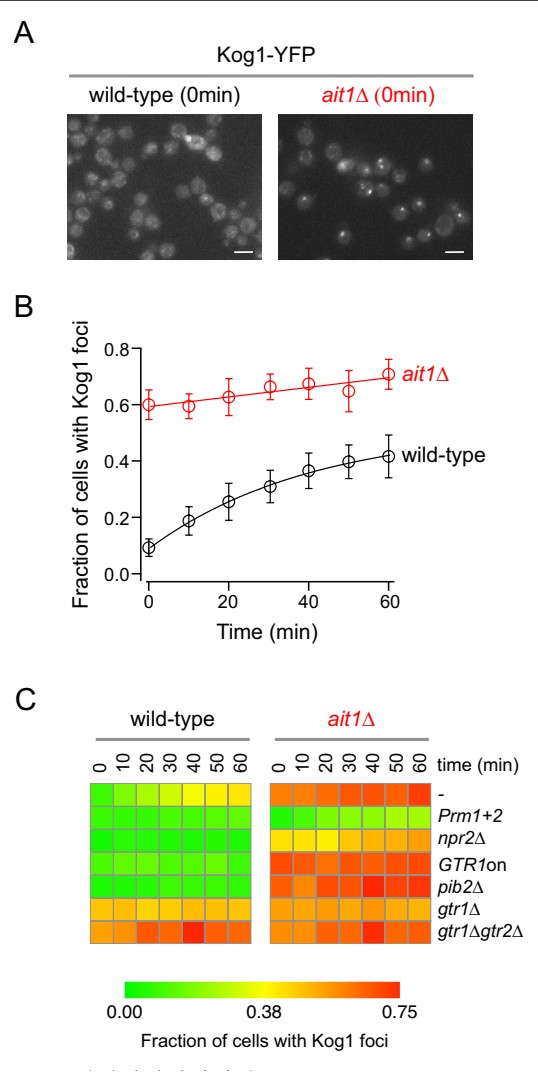

**Figure 5.** Target of rapamycin complex I (TORC1)-body formation in the *ait1Δ* strain. (**A**) Kog1-YFP localization in the wild-type and *ait1Δ* strains, during log-phase growth in nutrient-rich (SD) medium. The white bar shows 5 µm. (**B**) Fraction of wild-type and *ait1Δ* cells that form Kog1-YFP foci during nitrogen starvation. The points and error bars show the average and standard deviation from five replicate experiments, with at least 100 cells examined, per timepoint, per replicate. (**C**) Impact of the Ait1 deletion on TORC1-body formation in the wild-type and various mutant strains (compare left and right columns). Experiments were carried out in at least duplicate with over 200 cells examined per timepoint, per mutant. Individual timepoints have errors ranging from 0.05 to 0.10.

The online version of this article includes the following figure supplement(s) for figure 5:

**Figure supplement 1.** Gtr1 and Pib2 localization in the *ait1Δ* strain.

## Ait1 acts at, or above, the level of Gtr1/2 to regulate TORC1

The observation that Ait1 and Gtr1/2 both regulate TORC1 during amino acid starvation led us to consider two models of Ait1 function: (1) Ait1 acts at, or above, the level of Gtr1/2 to promote TORC1 inhibition, and (2) Ait1 acts downstream of Gtr1/2 to repress TORC1 activity once Gtr1/2 are inactivated. To distinguish between these models, we measured the impact that Ait1 has on TORC1 signaling in strains with: Gtr1 locked in its GDP-bound, inactive, state (GTR1$^{S20L}$ or Gtr1$^{off}$ for short); Gtr2 locked in its GTP-bound, inactive, state (GTR2$^{Q66L}$ or Gtr2$^{off}$ for short); and Gtr1 and Gtr2 both locked in their inactive states (Gtr1$^{off}$/Gtr2$^{off}$) (*Panchaud et al., 2013*). These experiments showed that Ait1 is still important for TORC1 inhibition in a Gtr1$^{off}$ strain, has limited impact on TORC1 inhibition in a Gtr2$^{off}$ strain, and actually helps activate TORC1 in a Gtr1$^{off}$/Gtr2$^{off}$ strain (*Figure 7*). Thus,

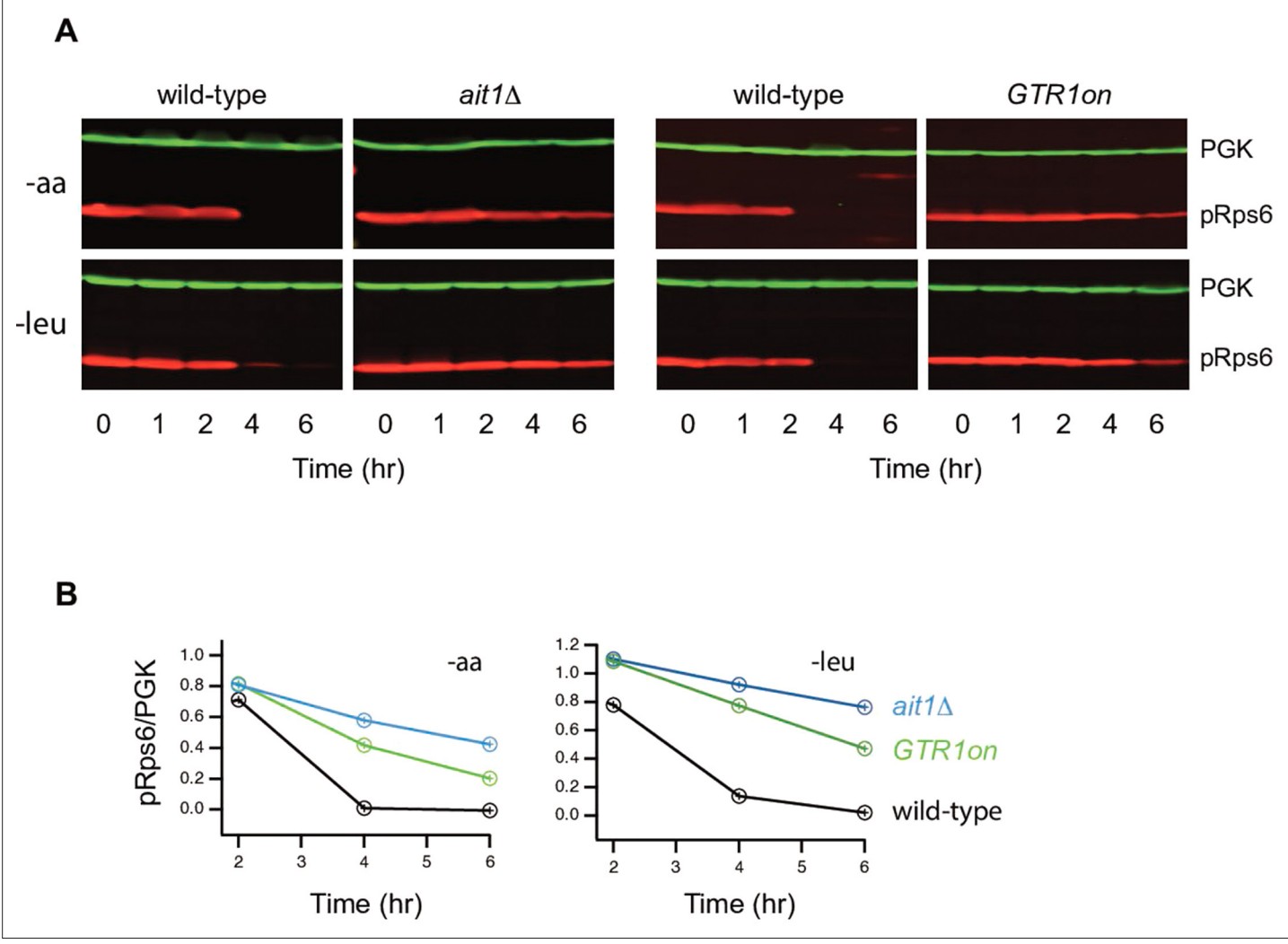

**Figure 6.** Impact of Ait1 on target of rapamycin complex I (TORC1) activity during amino acid starvation. (**B**) TORC1 activity during complete amino acid starvation (top), and leucine starvation (bottom), in wild-type, *ait1Δ*, and GTR1$^{Q65L}$ (Gtr1$^{on}$) strains, as measured by western blot using an anti phospho-Rps6 antibody. (**b**) Values show the ratio of the p-Rps6 signal divided by the PGK (loading control) signal in each lane, relative to the value for the wild-type strain at time = 0. Wild-type and mutant strains were grown and processed together and run on the same gel.

The online version of this article includes the following figure supplement(s) for figure 6:

**Figure supplement 1.** Impact of Ait1 on target of rapamycin complex I (TORC1) activity during nitrogen and amino acid starvation.

Ait1 promotes TORC1 repression via Gtr1/2 during amino acid starvation, likely by helping to drive Gtr2 into its inactive, GTP-bound, state. In line with this, deletion of Gtr1, Gtr2, or Gtr1/2 completely bypasses the need for Ait1 in amino acid starvation-dependent TORC1 signaling (*Figure 7*).

## Role of the C3 loop in Ait1-dependent TORC1 inhibition

To gain insight into the mechanism underlying Ait1-dependent regulation of Gtr1/2, we performed sequence alignments to look for similarity between Ait1 and known Gtr1/2- and RagA/C-binding proteins. These alignments uncovered analogous sequences in the N-terminal region of SLC38A9 and the third cytosolic (C3) loop of Ait1 (*Figure 8A*). The N-terminal region of SLC38A9 has been shown to dissociate from the pore of the SLC38A9 channel in the presence of arginine, and then bind (via residues 39–97; box *Figure 8A*) to a cleft at the interface between RagA and RagC (*Wang et al., 2015*; *Wyant et al., 2017*; *Lei et al., 2018*; *Fromm et al., 2020*). This cleft sits near the GTP-binding pockets in RagA and RagC and faces up and away from TORC1 in the RagA/C-TORC1 complex, at a distance ~80 Å away from the vacuolar surface/membrane (*Rogala et al., 2019*; *Fromm et al., 2020*).

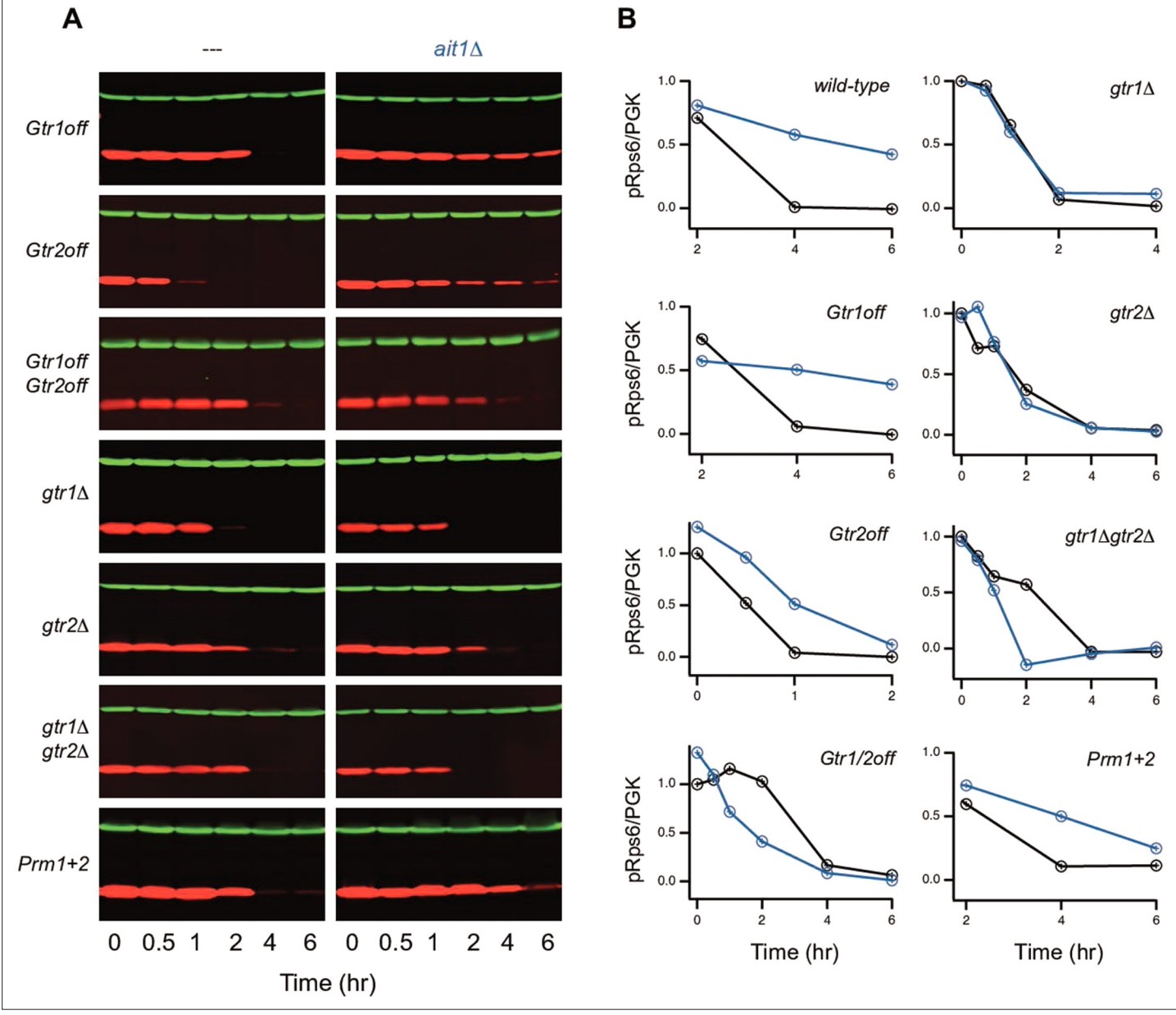

**Figure 7.** Impact of Ait1 on target of rapamycin complex I (TORC1) activity in strains with mutations in Gtr1/2, or the prion domains in Kog1, during amino acid starvation. (**A**) TORC1 activity during complete amino acid starvation in mutant strains with (left column) and without Ait1 (right column), measured using a western blot, as described in *Figure 6*. (**B**) Values show the ratio of the p-Rps6 signal divided by the PGK (loading control) signal in each lane, relative to the value for the wild-type strain at time = 0. Mutant and double mutant strains were grown and collected together and run on the same gel.

Thus, it seemed likely that Ait1 inhibits Gtr1/2, at least in part, via its 180 aa long, and intrinsically disordered, C3 loop. In line with this model, there are over 50 amino acids either side of the putative Gtr1/2-binding sequence in the C3 loop (box, *Figure 8A*)—more than enough unstructured peptide for the C3 loop to extend over TORC1 and interact with Gtr1/2.

To test If Ait1 regulates Gtr1/2 and TORC1 via its C3 loop, we first built two mutant versions of Ait1; one in which the C3 loop, and the other in which the C4 loop, is replaced by the short, flexible, linker GGSGSGEGSGSGG (*ait1Δc3* and *ait1Δc4*, respectively). Both mutant proteins fold and are trafficked to the vacuolar membrane, as judged by GFP-AitΔC3 and GFP-AitΔC4 localization (*Figure 8—figure supplement 1*). However, in line with the C3 loop model, only *aitΔc3* cells had a defect in TORC1 inhibition during leucine starvation (*Figure 8B, C*).

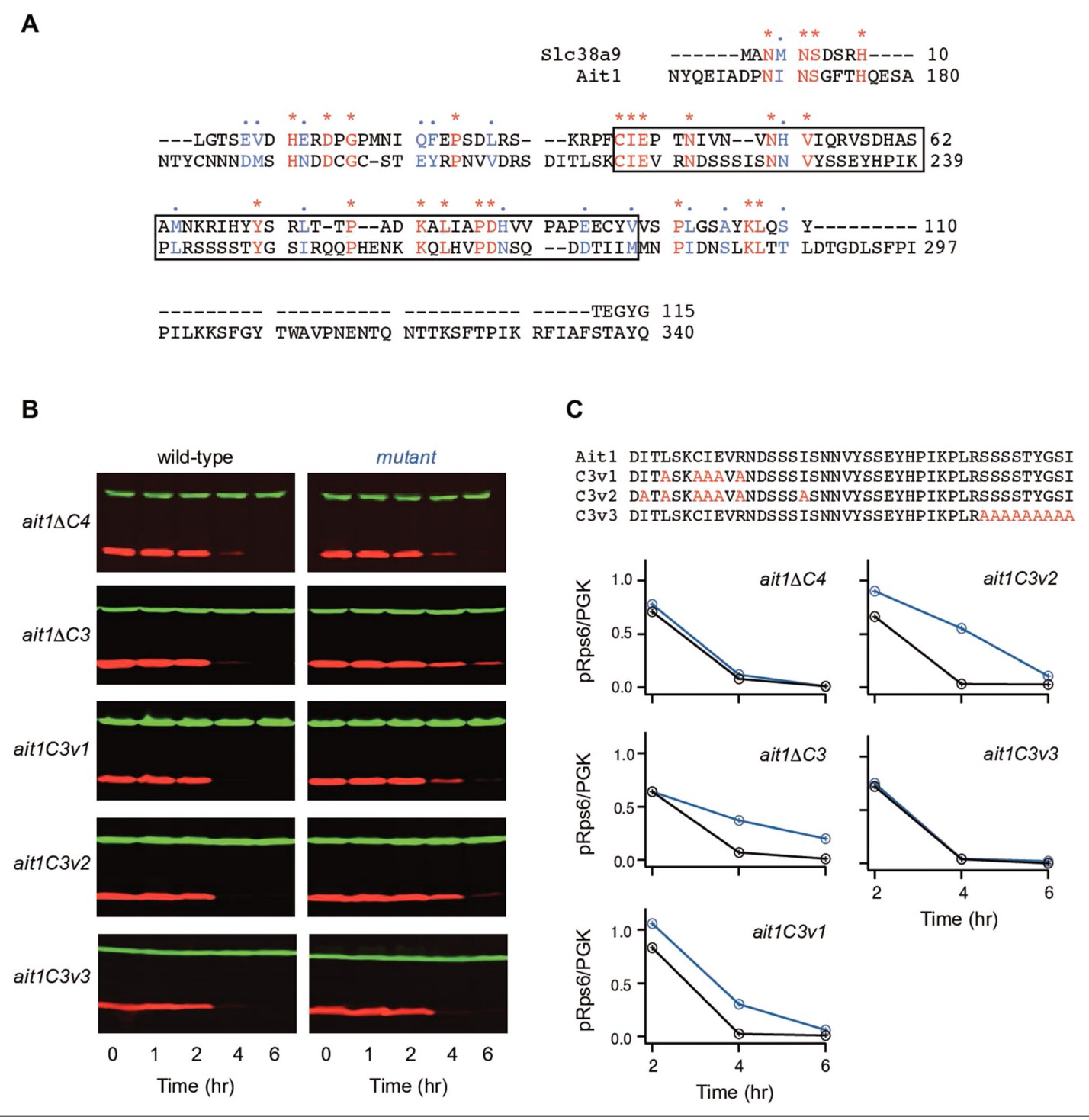

**Figure 8.** Role of the Ait1 C3 loop in target of rapamycin complex I (TORC1) regulation. (**A**) BLAST alignment of the SLC38A9 (top) and Ait1 sequences, showing the entire C3 loop (no other sequences in these proteins align). (**B, C**) TORC1 activity in Ait1 C3 and C4 loop mutants during leucine starvation, measured as described in *Figure 6*. Values show the ratio of the p-Rps6 signal divided by the PGK (loading control) signal in each lane, relative to the value for the wild-type strain at time = 0. Mutant and wild-type strains were grown and collected together and run on the same gel.

The online version of this article includes the following figure supplement(s) for figure 8:

**Figure supplement 1.** Localization of Ait1 and the Ait1 C3 and C4 loop mutants.

**Figure supplement 2.** Sequence alignments of Ait1 from a variety of yeast species, showing the high degree of conservation at the center of the C3 loop.

**Figure supplement 3.** Condition-dependent signaling changes in Ait1C3v2.

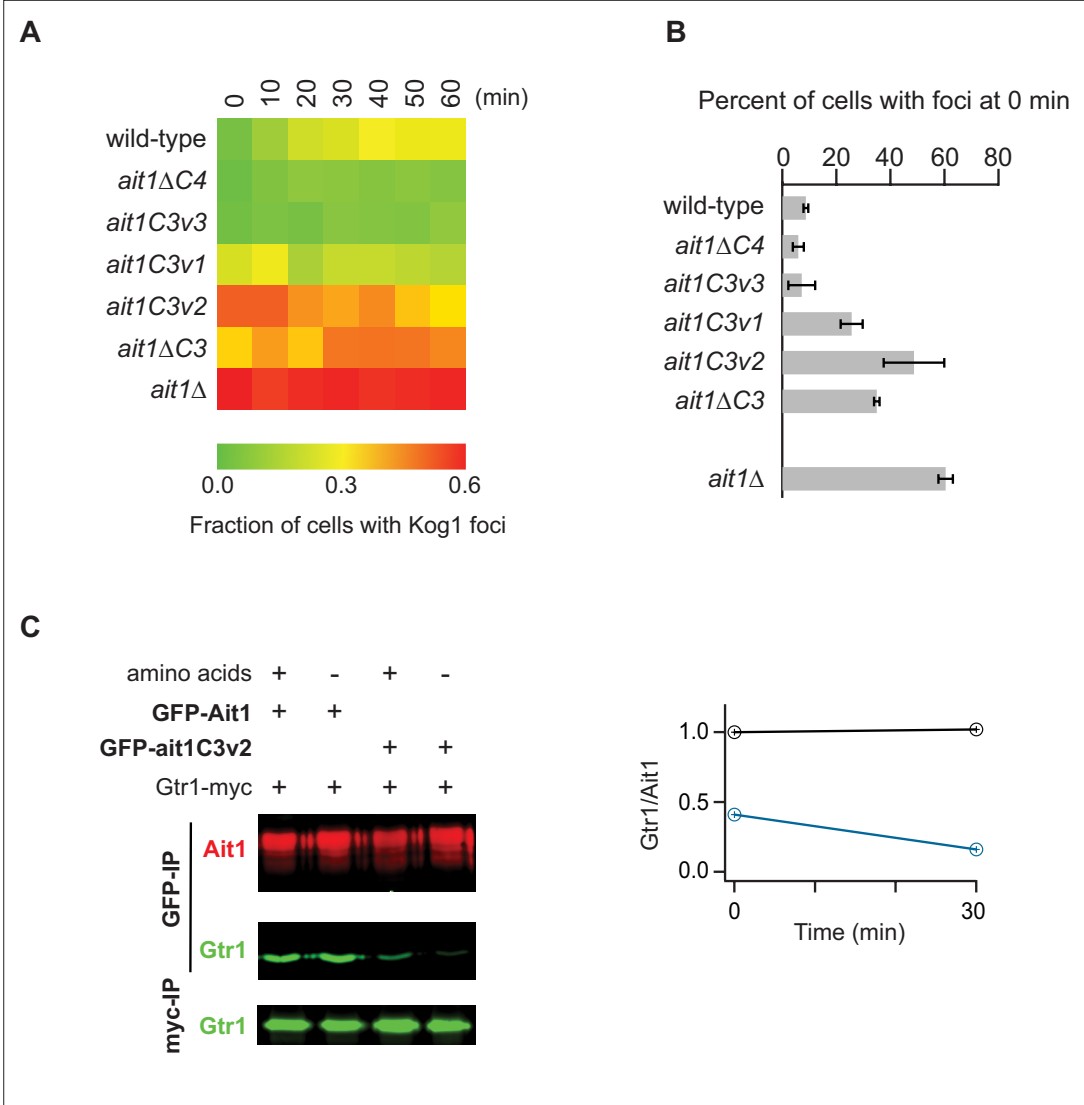

**Figure 9.** Impact of Ait1 C3 and C4 loop mutations on target of rapamycin complex I (TORC1) localization and Gtr1/2 binding. (**A**) Each square on the heat map shows the fraction of cells with a Kog1-YFP focus/body at a specific timepoint (as labeled), calculated by averaging the data from three replicate experiments (>100 cells analyzed at each timepoint and replicate). Individual timepoints have errors ranging from 0.02 to 0.12 (average 0.06). (**B**) Bar graph showing the fraction of cells with a TORC1 body during log-phase growth in SD medium. (**C**) Coimmunoprecipitation showing a strong interaction between GFP-Ait1 and Gtr1-myc, but not GFP-Ait1C3v2 and Gtr1-myc. The graph shows the ratio of the Gtr1 and Ait1 signals in the wild-type (black line) and Ait1C3v2 (blue line) strains, before, and 30 min after, amino acid starvation. The Immunoprecipitation data with a full set of controls are shown in *Figure 9—figure supplement 1*.

The online version of this article includes the following figure supplement(s) for figure 9:

**Figure supplement 1.** Coimmunoprecipitation of GFP-Ait1 and Gtr1-myc.

Next, to test the function of the central portion of the C3 loop, we created two strains carrying mutations in the region running from Arg 208 to Tyr 231 since it is highly conserved across the yeasts (*Figure 8—figure supplement 2*): In the first strain (*ait1c3v1*) we mutated five residues at the center of the 208–231 stretch to alanine (L213A, C216A, I217A, E218A, and R220A; *Figure 8C*). In the second strain (*ait1c3v2*), we added two mutations to *ait1c3v1* (I211A and I226A; *Figure 8C*). We also created a control strain (*ait1c3v3*) that has nine mutations in a poorly conserved portion of the loop (residues 242–250; *Figure 8C*). All three mutant proteins fold and are transported to the vacuolar membrane as judged by GFP-Ait1 localization (*Figure 8—figure supplement 1*).

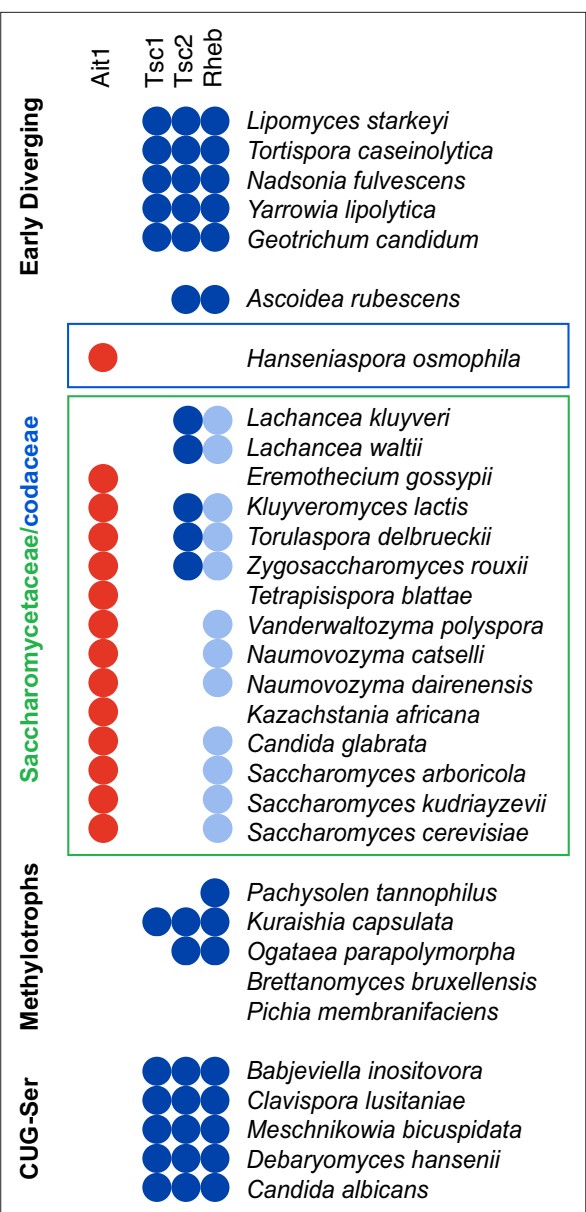

**Figure 10.** Evolution of the target of rapamycin complex I (TORC1) circuit in yeast. Species identified as carrying Ait1 in a BLAST search (p < 0.001 cutoff) are marked with a red circle on a previously constructed map of Rheb and TSC1/2 conservation among the budding yeast, taken from **Tatebe and Shiozaki, 2017**. The light blue circles denote the presence of a highly divergent (nonfunctional) Rheb in species closely related to *S. cerevisiae* (**Tatebe and Shiozaki, 2017**). Ait1 was not detected in any of the yeasts outside the *Saccharomycetaceae* and *Saccharomycodaceae*.

As expected, *ait1c3v1* and *ait1c3v2* cells had defects in TORC1 inhibition during leucine and amino acid starvation—a phenotype not seen in the *ait1c3v3* strain (**Figure 8B, C**, **Figure 8—figure supplement 3**). Importantly, however, the signaling defect was larger in *ait1c3v2* cells than in *ait1c3v1* cells, indicating that Ile 211 and/or Ile 226 play a key role in the Ait1-dependent regulation of TORC1 (**Figure 8B, C**). Disruption of the C3 loop also triggered the formation of TORC1 bodies. Specifically, the *ait1c3v1*, *ait1c3v2*, and *aitΔc3* strains all form more TORC1 foci during nitrogen starvation, and/or log-phase growth, than the wild-type strain (**Figure 9A, B**). Again here, *ait1c3v2* cells had a stronger phenotype than *ait1c3v1* cells, consistent with the idea that the mutations in Ait1C3v1 only partially disrupt the C3 loop.

Finally, to see if the C3 loop drives an interaction with Gtr1/2, we immunopurified GFP-Ait1 and GFP-AitC3v2 in nonionic detergent (without a crosslinker), in the presence and absence of amino acids, and then measured the amount of Gtr1-myc that remains associated with Ait1. This experiment showed (1) that Ait1 interacts with Gtr1/2 in both log-phase growth and starvation conditions, and (2) that the Gtr1/2-Ait1 interaction is significantly weaker in the *ait1c3v2* strain (*Figure 9C*, *Figure 9— figure supplement 1*).

Together, these data show that the C3 loop in Ait1 helps hold TORC1 in place around the vacuolar membrane, and drives TORC1 inhibition during amino acid starvation, likely by binding directly to Gtr1/2. We note, however, that the signaling defects seen in the *ait1c3v2* and *aitΔc3* strains (*Figure 8* and *Figure 8—figure supplement 3*) are smaller than those seen in the *ait1Δ* strain (*Figure 6*), suggesting that Ait1 also regulates TORC1 via regions outside of its C3 loop.

## Discussion

In this report, we show that: (1) The previously unstudied GPCR-like protein, Ait1, binds directly or indirectly to TORC1-Gtr1/2 and holds TORC1 around the vacuolar membrane during log-phase growth; (2) Ait1 acts through Gtr1/2 (most likely Gtr2) to inhibit TORC1 during amino acid starvation; and (3) Ait1 regulates TORC1, in part, via a central region in its 180 amino acid C3 loop that resembles the Rag A/C (Gtr1/2)-binding domain in SLC38A9.

Putting our results together with the previously published work showing that the GAP SEAC inactivates Gtr1 during nitrogen and amino acid starvation (*BarPeled et al., 2013*; *Panchaud et al., 2013*; *Neklesa and Davis, 2009*; *Chen et al., 2017*; *Laxman et al., 2014*; *Algret et al., 2014*), we propose the following model: First, at the onset of amino acid starvation, SEAC is activated and triggers GTP hydrolysis in Gtr1. The resulting conformational change in Gtr1, and/or signals transmitted through Ait1, then trigger a change in Ait1 structure/activity. Next, Ait1 drives the release of GDP from Gtr2, completing the conversion of Gtr1/2 from the active GTP/GDP-bound state, to the inactive GDP/GTP-bound state. Finally, the resulting conformational change in Gtr1/2, and subsequent movement of the C3 loop in Ait1, exposes a key surface on TORC1 to initiate TORC1-body formation.

As an extension of this model, it may be that strong signals through SEAC, such as those in complete nitrogen starvation (*Figure 6—figure supplement 1*), override the need for Ait1 due to coupling between the GTP-binding sites in Gtr1 and Gtr2 (*Shen et al., 2017*).

It is important to point out, however, that while our working model provides a simple explanation for all of the available data (including the observation that deletion/mutation of Ait1 triggers TORC1-body formation), it remains unclear how Ait1 regulates Gtr1/2 and TORC1; Ait1 could also promote TORC1 inhibition by stabilizing the inactive state of Gtr1/2, controlling access to Gtr1/2 activators/ repressors (including SEAC), and/or other mechanisms.

The dual role of Ait1 in holding TORC1 around the vacuolar membrane and helping to regulate TORC1 via Gtr1/2 is especially interesting in the context of yeast evolution. Ait1 is found in species throughout the closely related *Saccharomycetaceae* and *Saccharomycodaceae*, but not in other clades (*Supplementary file 1*). The *Saccharomycetaceae/codaceae*, which include *S. cerevisiae*, *Ashbya gossypii*, *Kluyveromyces lactis*, and the pathogen *C. glabrata*, split from other yeasts approximately 200 million years ago (*Shen et al., 2018*), and are unique in that they have highly divergent Rheb, or no Rheb, and have lost TSC2 and/or TSC1 (*Figure 10*). They are also unique in that many species in these families have prion-like, glutamine-rich, domains in Kog1/Raptor; domains that at least in *S. cerevisiae* help control the commitment to quiescence (*Hughes Hallett et al., 2015*). Thus, a common ancestor of the *Saccharomycetaceae/codaceae* lost functional Rheb and TSC1/2, gained prion-like domains in Kog1, and gained Ait1.

We argue that these events were almost certainly linked. First, the data presented here show that Ait1 is required to block TORC1 from agglomerating via its prion-like domains during log-phase growth in nutrient replete conditions. That is, without Ait1-dependent anchoring of TORC1, the addition of prion-like domains in TORC1 would have triggered constitutive body formation, rather than add a reversible, regulatory, transition to the TORC1 circuit. Second, our data show that Ait1 has taken over part of the role of ancestral Rheb (*Fukuda et al., 2021*) by helping to regulate TORC1 in starvation conditions.

Further work is needed to fully dissect the structure and function of the TORC1 circuit in *S. cerevisiae*, and other simple eukaryotes, and explore the implications of the rewiring we have uncovered.

For example, it still remains uncertain how Gtr1/2 regulate TORC1 in the absence of Rheb. However, what is clear from this, and other recent studies, is that while the core of the TORC1 pathway (including TORC1 itself and Gtr1/2) is highly conserved, other aspects of this ancient growth control circuit are plastic and have changed significantly during evolution. Learning more about these changes will not only shed light on the way that eukaryotes have adapted to different niches, it should also open the door to creating drugs that specifically block the growth of fungal pathogens and a variety of parasites.

## Materials and methods

### Strain construction

All strains used in this study were generated in haploid (W303) *S. cerevisiae*, using standard methods (*Storici and Resnick, 2006*; *Storici et al., 2001*), and are listed in *Supplementary file 3*.

### Crosslinking and immunopurification

Yeast carrying Kog1-FLAG, and separately Kog1-HA, were inoculated into 5 ml of synthetic complete medium containing 2% glucose (SD medium) and grown overnight at 30°C in a 20 ml tube, rotating at 40 rpm. The cells were then: (1) diluted to an $OD_{600}$ of 0.1 in 250 ml of fresh SD medium, and grown shaking at 200 rpm and 30°C in a 1 l flask, until they reached an $OD_{600}$ of 0.6; (2) captured by filtration, washed with 2 × 100 ml of the appropriate stress or starvation medium, and transferred into 200 ml of synthetic medium lacking all nitrogen (-N), all glucose (-Glu), SD medium containing 0.4 M KCl or 1 mM $H_2O_2$, or SD medium at 42°C; (3) grown again for the indicated period of time (*Figure 1*), shaking at 200 rpm and 30°C (or 42°C for heat stress) in a 1 l flask; (4) harvested by filtration, and rinsed into 2 ml screw-cap tubes using a small volume of ImmunoPrecipitation Lysis Buffer (IPLB; 20 mM 4-(2-hydroxyethyl)-1-piperazineethanesulfonic acid (HEPES), pH 7.5, 150 mM potassium acetate, 2 mM magnesium acetate, 1 mM ethylene glycol bis(2-aminoethyl)tetraacetic acid (EGTA), and 0.6 M sorbitol) (*Murley et al., 2017*); and (5) centrifuged for 30 s at 8000 rpm, the supernatant discarded, and the pellet flash frozen, and stored at −80°C.

To lyse the cells, the frozen pellets were resuspended in approximately 600 µl of IPLB buffer containing protease and phosphatase inhibitors (Roche, Indianapolis, IN; 04693159001 and 04906845001; IPLB⁺⁺), and 1 ml of glass microbeads, and the slurries subjected to 6 × 1 min of vigorous shaking in a Mini-Beadbeater-24 (BioSpec) at 4°C. The tubes were then punctured using a 23-gauge needle and the lysates eluted into 1.5 ml tubes by centrifugation at 3000 rpm at 4°C, for 5 min. The lysates were then homogenized by gentle vortexing, decanted into a fresh 1.5 ml tube, and treated with 0.25 µM of DSP at 4°C for 30 min (with gentle rotation). At this point crosslinking was then quenched by adding 70 µl of 1 M Tris–HCl, pH 7.5, to each tube and holding the extracts on ice for 30 min. Finally, 1% digitonin was added to each tube, and the extracts incubated at 4°C for 1 hr (with gentle rotation), before they were clarified by centrifugation at 12,000 rpm at 4°C, for 10 min, and the supernatant transferred into a fresh tube.

To purify Kog1 and any crosslinked interactors, 50 µl of µMACS anti-FLAG beads (Miltenyi Biotech, 130-101-591) was added to each clarified extract, and the tubes rotated at 4°C for 1.5 hr. The µMACS columns were then prepared by washing them with 200 µl of the lysis buffer supplied with the purification kit, followed by 200 µl of IPLB⁺⁺ containing 1% digitonin, before the bead/extract mix was loaded into each column (on a magnet) and allowed to flow through by gravity. The beads were then washed in three steps: (1) four times with 200 µl of IPLB⁺⁺ containing 0.1% digitonin, (2) two times with 400 µl of IPLB⁺⁺ containing no digitonin, and (3) once with 200 µl of 20 mM Tris–HCl, pH 7.5. Kog1 and any crosslinked proteins were then eluted by incubating each column with 20 µl of the elution buffer supplied with the kit (heated to 95°C), for 5 min, and then adding of 2 × 40 µl of the same elution buffer containing 50 mM DL-Dithiothreitol (DTT;also at 95°C). The pooled eluate from each column was then loaded into a single lane on a 10% sodium dodecyl sulfate (SDS)–polyacrylamide gel and allowed to migrate until it completely entered the gel. The gels were then stained with colloidal blue, destained, and the lane excised for analysis by mass spectrometry.

Identical procedures were used to identify Pib2 and Ait1 interactors, except that in the these experiments the IP was done using GFP-Pib2 or GFP-Ait1 and anti-GFP beads (Miltenyi Biotech, 130-101-125).

## Protein identification by mass spectrometry

Gel slices were washed with water, 50% acetonitrile/50% water, acetonitrile, ammonium bicarbonate (100 mM), and then 50% acetonitrile/50% ammonium bicarbonate (100 mM). The liquid was then removed from each sample, and the gel slices dried in a speed vac. The gel slices were then: (1) Reduced with dithiothreitol (10 mM in 100 mM ammonium bicarbonate) at 56°C for 45 min, and the solution removed and discarded. (2) Alkylated with iodoacetamide (55 mM in 100 mM ammonium bicarbonate) in the dark at ambient temperature for 30 min. (3) Washed with ammonium bicarbonate (100 mM) for 10 min on a shaker, an equal volume of acetonitrile added, and washed for an additional 10 min on a shaker, and then dried in a speed vac for 45 min. (4) Cooled on ice and a treated with a cold solution of 12.5 ng/µl trypsin (Promega, Madison, WI) in ammonium bicarbonate (100 mM). After 45 min, the trypsin solution was removed and discarded, and an equal amount of ammonium bicarbonate (50 mM) was added, and the sample incubated overnight at 37°C with mixing. The samples were then spun down in a microfuge and the supernatants collected. Peptides were further extracted from the gel slices by adding 0.1% trifluoroacetic acid (TFA; enough to cover the gel slices) and mixed at ambient temperature for 30 min. An equal amount of acetonitrile was then added, and the samples were mixed for an additional 30 min. The samples were then spun on a microfuge and the supernatants pooled and concentrated in a speed vac. Finally, all samples were desalted using ZipTip $C_{18}$ (Millipore, Billerica, MA) and eluted with 70% acetonitrile/0.1% TFA, and concentrated in a speed vac.

For analysis, the peptide samples were brought up in 2% acetonitrile and 0.1% formic acid (10 µl) and analyzed (8 µl) by LC/ESI MS/MS with a Thermo Scientific Easy1000 nLC (Thermo Scientific, Waltham, MA) coupled to a hybrid Orbitrap Fusion (Thermo Scientific, Waltham, MA) mass spectrometer. Inline desalting was accomplished using a reversed-phase trap column (100 µm × 20 mm) packed with Magic $C_{18}$AQ (5 µm 200 Å resin; Michrom Bioresources, Auburn, CA) followed by peptide separations on a reversed-phase column (75 µm × 250 mm) packed with Magic $C_{18}$AQ (5 µm 100 Å resin; Michrom Bioresources, Auburn, CA) directly mounted on the electrospray ion source. A 90 min gradient from 2% to 35% acetonitrile in 0.1% formic acid at a flow rate of 300 nl/min was used for chromatographic separations. A spray voltage of 2000 V was applied to the electrospray tip and the Orbitrap Fusion instrument was operated in the data-dependent mode, MS survey scans were in the Orbitrap (AGC target value 500,000, resolution 120,000, and injection time 50 ms) with a 3-s cycle time and MS/MS spectra acquisition were detected in the linear ion trap (AGC target value of 10,000 and injection time 35 ms) using HCD activation with a normalized collision energy of 27%. Selected ions were dynamically excluded for 45 s after a repeat count of 1.

Data analysis was performed using Proteome Discoverer 2.2 (Thermo Scientific, San Jose, CA). The data were searched against an SGD yeast database that included common contaminants. Searches were performed with settings for the proteolytic enzyme trypsin. Maximum missed cleavages were set to 2. The precursor ion tolerance was set to 10 ppm and the fragment ion tolerance was set to 0.6 Da. Variable modifications included oxidation on methionine (+15.995 Da) and carbamidomethyl (57.021). Sequest HT was used for database searching. All search results were run through Percolator for scoring.

## Fluorescence microscopy

TORC1-body formation was measured as described previously (*Hughes Hallett et al., 2015*; *Sullivan et al., 2019*). Briefly, stains carrying Kog1-YFP were patched from their glycerol stocks onto fresh YEPD plates and grown overnight at 30°C. The patches were then used to inoculate 5 ml of SD medium, and the tubes grown at 30°C in a 20 ml tube, rotating at 40 rpm, until they reached an $OD_{600}$ of 0.1. These starter cultures were then used to inoculate 20 ml of SD medium in a 150-ml Erlenmeyer flask (to an $OD_{600}$ below 0.01) and grown at 30°C and shaking at 200 rpm, until they reached an and $OD_{600}$ between 0.5 and 0.7. 300 µl of each culture was then pipetted into one chamber in an 8-well micro-slide (Ibidi, 80826) that had been pretreated with concanavalin A. The chambers were then washed three times with SD -nitrogen, and images acquired using a Nikon Eclipse Ti-E microscope equipped with a ×100 objective, a Photometrics Prime 95B camera, and $\lambda_{EX}$ 510/25 and $\lambda_{EM}$ 540/21 filters, every 10 min for an hour. Each image consisted of a z-stack of sixteen 200 ms images, spaced 0.4 µm apart, to ensure that the bodies in all planes were detected, and was compressed into a maximum projection stack in ImageJ for analysis.

Imaging of GFP-Ait1, Gtr1-YFP, and GFP-Pib2 was done in an identical manner except that GFP images were acquired $\lambda_{EX}$ 470 and $\lambda_{EM}$ 515/30 filters.

## Rps6 phosphorylation assays

Cultures were grown in conical flasks shaking at 200 rpm and 30°C until mid-log phase (OD$_{600}$ between 0.55 and 0.6). At this point, a 47 ml sample was collected, mixed with 3 ml 100% trichloroacetic acid (TCA), and held on ice for at least 30 min (and up to 6 hr). The remaining culture was then collected by filtration, and transferred to SD -N, SD -aa, or SD -leu medium after two washes with 100 ml of the same medium, and further samples collected in TCA, as described above. The samples were then centrifuged at 4000 rpm for 5 min at 4°C, washed twice with 4°C water, twice with acetone, and disrupted by sonication at 15% amplitude for 5 s before centrifugation at 8000 rpm for 30 s. The cell pellets were then dried in a speedvac for 10 min at room temperature, and frozen until required at −80°C.

Protein extraction was performed by bead beating (6 × 1 min, full speed) in urea buffer (6 M urea, 50 mM Tris–HCl pH 7.5, 5 mM Ethylenediaminetetraacetic acid (EDTA), 1 mM phenylmethylsulfonyl fluoride (PMSF), 5 mM NaF, 5 mM NaN$_3$, 5 mM NaH$_2$PO$_4$, 5 mM $p$-nitrophenylphosphate, 5 mM β-glycerophosphate, and 1% SDS) supplemented with complete protease and phosphatase inhibitor tablets (Roche, Indianapolis, IN; 04693159001 and 04906845001). The lysate was then harvested by centrifugation for 5 min at 3000 rpm, resuspended into a homogenous slurry, and heated at 65°C for 10 min. The soluble proteins were then separated from insoluble cell debris by centrifugation at 12,000 rpm for 10 min, and the lysate stored at −80°C until required.

For protein phosphorylation analysis, the protein extracts were run on a 12% acrylamide gel and transferred to a nitrocellulose membrane. Western blotting was then carried out using anti-pRPS6 antibody (Cell Signaling, 4858) at a 1/2500 dilution, and anti-PGK1 antibody (Invitrogen, 459250) at a 1/10,000 dilution, and anti-mouse and anti-rabbit secondaries, labeled with a IRDye 700CW and IRDye 800CW (LiCor), both at a 1/10,000 dilution, and the blots scanned using a LiCor Odyssey Scanner (LiCor, Lincoln, NE). Band intensities were quantified using the LiCor Image Studio Software.

## Sch9 bandshift experiments

Sch9 bandshift measurements were performed as described previously (*Urban et al., 2007*; *Hughes Hallett et al., 2014*), and using the same procedure listed above for the Rps6 Western, except that lysates were subjected to cleavage by 2-nitro-5-thiocyanatobenzoic acid (NTCB) for 12–16 hr at room temperature (1 mM NTCB and 100 mM N-Cyclohexyl-2-aminoethanesulfonic acid (CHES, pH 10.5)) prior to analysis, and the Western was done using an anti-HA (12CA5) antibody.

## Coimmunoprecipitation experiments

*Figure 3—figure supplement 1*. Kog1-FLAG was immunopurified, as described above, the eluate run on a 9% SDS–polyacrylamide gel and transferred to a nitrocellulose membrane. Western blotting was then carried out using a rabbit Anti-GFP polyclonal antibody (enQuirebio, Cat# QAB10298) at a 1:2500 dilution, a mouse Anti-FLAG monoclonal antibody (Sigma Cat# F1804) at a 1:1000 dilution, and the same secondary antibodies used in the Rps6 assay. A sample of each extract (collected prior to immunopurification) was also run on a separate 12% gel and probed with anti-PGK1 antibody (Invitrogen, 459250) at a 1/10,000 dilution, as described in the Rps6 assay. Ait1 levels were monitored in the same cell lines and conditions, but after immunopurification (following the protocol described below).

*Figure 9* and *Figure 9—figure supplement 1*. GFP-Ait1 was immunopurified as described above for Kog1-Flag, but with a few modifications: (1) no crosslinker was used; (2) the beads were washed three times with 200 µl of IPLB$^{++}$ containing 0.1% digitonin, and once with 400 µl of IPLB$^{++}$ containing no digitonin; (3) Anti-GFP MicroBeads (Miltenyi Biotec Cat# 130-091-125) were used instead of Anti-FLAG beads. Once the immunopurification was complete, the eluted material was run on a 12% SDS–polyacrylamide gel and transferred to a nitrocellulose membrane. Western blotting was then carried out using a rabbit Anti-GFP polyclonal (enQuirebio, Cat# QAB10298) at a 1:2500 dilution, a mouse Anti-Myc (9E10) (Roche Cat# 1257900), and the same secondary antibodies listed in the Rps6 assay. Gtr1-myc was immunoprecipitated using an anti-Myc antibody (Roche cat# 1257900) and protein A/G beads (Santa Cruz cat# sc-2003).

## Acknowledgements

We thank Claudio De Virgilio for sharing GTR1 and 2 mutant plasmids, and Kyle Cunningham for sharing the GFP-Pib2 plasmid, used to make mutant strains. We also thank Phil Gafken and Lisa Jones of the Fred Hutchinson Cancer Research Center's Proteomics Facility, and Paul Langlais at the University of Arizona for carrying out the peptide mapping experiments. This work was supported by the National Institutes of Health (NIH) grants R01GM097329 and T32GM136536.

## Additional information

### Funding

| Funder | Grant reference number | Author |
| --- | --- | --- |
| National Institute of General Medical Sciences | R01GM097329 | Andrew P Capaldi |
| National Institute of General Medical Sciences | T32GM136536 | Ryan L Wallace Andrew P Capaldi |
| University of Arizona | | Andrew P Capaldi |

The funders had no role in study design, data collection, and interpretation, or the decision to submit the work for publication.

### Author contributions

Ryan L Wallace, Data curation, Formal analysis, Investigation, Methodology, Writing - original draft, Writing - review and editing; Eric Lu, Conceptualization, Formal analysis, Investigation, Methodology, Writing - original draft, Writing - review and editing; Xiangxia Luo, Investigation, Methodology; Andrew P Capaldi, Conceptualization, Data curation, Formal analysis, Supervision, Funding acquisition, Investigation, Methodology, Writing - original draft, Project administration, Writing - review and editing

### Author ORCIDs

Eric Lu http://orcid.org/0000-0002-3144-3563
Andrew P Capaldi http://orcid.org/0000-0002-7902-2477

### Decision letter and Author response

Decision letter https://doi.org/10.7554/eLife.68773.sa1
Author response https://doi.org/10.7554/eLife.68773.sa2

## Additional files

### Supplementary files

• Supplementary file 1. Complete data for the Kog1 and Pib2 immunopurifications. The IP-mock tab shows the number of background corrected peptide spectral counts (PSMs) for each protein detected (rows) across the different IPs (columns). The Filterpass tab shows the number of raw PSMs in each IP and mock IP. Proteins with at least twofold higher abundance in the true IP (Kog1-FLAG or GFP-Pib2) versus the mock IP (Kog1-HA or wild-type Pib2), and with at least seven peptide spectral maps in the true IP, were scored as potential interactors.

• Supplementary file 2. Results of a BLASTP search against Ydl180w from *S. cerevisiae*. Subspecies/variants were removed from the table for clarity. No other families of yeast have species with significant *E* values (p < 0.01).

• Supplementary file 3. List of strains used in this study.

• Transparent reporting form

• Source data 1. Labelled raw gel images. The original labeled gel, from each panel in *Figure 3—figure supplement 1*, *Figure 6*, *Figure 6—figure supplement 1*, *Figure 7*, *Figure 8*, *Figure 8—figure supplement 3*, *Figure 9*, and *Figure 9—figure supplement 1* are included in source data in two separate folders. In each case the gels are numbered as they are shown in the associated

figure—from top to bottom. In the case of *Figure 6*, the two gels on the left are labeled 1 and 2 and the two gels on the right are labeled 3 and 4.

• Source data 2. Labelled raw gel images.

## Data availability

All data generated or analyzed during this study are included in the manuscript and supporting files.

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
