## [Editor Report]

Genetic analysis of the yeast *S. cerevisiae* identifies a role for the vacuolar GPCR Ait1 in the regulation of TORC1. The mechanism of Ait1 function is mediated by regulation of the small GTPases Gtr1/2. This finding identifies novel signaling modality in yeast for the control of TORC1 function.

---

## [Decision Letter]

**Decision letter after peer review:**

Thank you for submitting your article "Ait1 regulates TORC1 signaling and localization in budding yeast" for consideration by *eLife*. Your article has been reviewed by 3 peer reviewers, one of whom is a member of our Board of Reviewing Editors, and the evaluation has been overseen by Philip Cole as the Senior Editor. The reviewers have opted to remain anonymous.

This study uses a proteomics approach to identify proteins that interact directly or indirectly with the Torc1 component Kog1 and microscopy to determine whether these proteins influence the formation of TORC1 bodies, which have been established previously to form on vacuoles in budding yeast upon nitrogen starvation. The authors focus on an interesting new GPCR-like protein identified that they call Ait1, which when deleted results in constitutive formation of TORC1 bodies concomitant with resistance of TORC1 to suppression upon amino acid starvation. Genetic approaches suggest that Ait1 acts upstream of the Gtr1/Gtr2 proteins, known to directly engage TORC1 at the vacuole, for TORC1 suppression and that this suppression might involve sequences within a cytosolic loop of Ait1. The data are clear that Ait1 represents an important new component of TORC1 regulation by amino acids in *S. cerevisiae* that is not present in most eukaryotic species. The authors make rather specific conclusions regarding the function of Ait1 in the regulation of Gtr1/2 and TORC1. However, its molecular function(s) is incompletely characterized in the study, leaving the conclusions supported by ambiguous data, with several fundamental questions unanswered regarding its placement in the known regulatory network upstream of yeast TORC1.

Essential revisions:

Revisions required to address the stated conclusions:

1) At the heart of the authors' conclusions on the function of Ait1 is its genetic placement amongst core regulators of TORC1, especially the highly conserved Gtr1/2 proteins. Based on Figure 7, the authors conclude that Ait1 acts upstream of Gtr1/2 for TORC1 regulation (Gtr1/2 are genetically epistatic to Ait1). However, the data in Figure 5 clearly show that the ability of the Gtr1-GTP (GTR1on) and npr2-δ mutants to inhibit TORC1-body formation during nitrogen starvation is dependent on Ait1, thereby placing Ait1 downstream of Gtr1/2 for this regulation.

2) Page 5-6. The authors suggest that there is a direct interaction between Ait1 and TORC1, citing the evidence that they mutually pull down each other in IP-MS experiments. The following experiments are required to support this conclusion: a) Identify residues or mapping critical domain/motif(s) on Ait1 that required for interaction with TORC1; b) Native gel or gel-filtration analysis to show the co-migration of Ait1 and TORC1; c) in vitro binding assay with purified proteins is preferred and thus recommended; d) Co-IP experiment to show the interaction between Ait1 and TORC1 is regulated by nitrogen/amino acid availability.

3). Figure 8C describes a key experiment suggesting that the ait1C3v1 mutant is a potential separation-of-function mutant, defective in suppressing TORC1 upon leucine starvation but maintaining its ability to prevent TORC1 body formation. However, this effect is never properly examined under nutrient deprivation time courses. Kog1-YFP foci formation over a time course of nitrogen, leucine, and amino acid starvation in this mutant versus wild-type and ait1-δ should be quantified to better demonstrate the effects of this mutant on these two putative functions of Ait1. Effects on pRps6 under nitrogen and amino acid starvation should also be shown.

4) Page 8: The authors suggest that Ait1 and SLC38A9 share sequence homology so they likely bind to Rag/Gtr in a similar way. The following experiments are needed to support this claim: a) A critical motif in SLC38A9 that mediates the binding with the Rags is an α helix (H60-Y71). In the sequence alignment of Figure 8A, the corresponding region on Ait1 contains no feature of an α helix because multiple serine residues will likely introduce flexibility to prevent α helix formation. The authors show provide stronger evidence that this loop directly contacts the Rags; b) Based on the cryo-EM structure of SLC38A9-Rag, the triple mutant that the authors generated (Ait1c3v1) is not appropriate. None of the three residues directly contact the Rags in the structure, so its defect is difficult to explain; c) What is the molecular mechanism of Ait1 function? Does it change the GTP/GDP state of Gtr? Is it a GEF, GDI, or GAP?

5) What signal does Ait1 respond to? Does it bind amino acids? Does Ait1 mediate the effects of leucine or other amino acids?

In the possible absence of the requested studies, the conclusions presented should be modified to include only those that are directly supported by experiment.

*Reviewer #1:*

This is an interesting study that identifies a novel component of the amino acid signaling pathway that regulates TORC1 activity in yeast. It is proposed that the 3rd cytoplasmic loop of GPCR Ait1 recruits TORC1 to the lysosomal surface in response to amino acids by a mechanism analogous to SLC38A9 in humans. The importance of this study is that it highlights the rewiring of the TORC1 signaling pathway in yeast compared to humans.

The major strength of this study is the rigor of the mass spec analysis used to identify novel TORC1 signaling components coupled with yeast genetic analysis. This leads to high confidence that the conclusions of this study are correct. However, there are a number of key issues that are not addressed by the authors, including the direct demonstration of a role for amino acids in Ait1 function. The study will be appropriate for publication in *eLife* after these issues have been addressed in a revised manuscript.

*Reviewer #2:*

This paper identified a potentially key mediator of nitrogen/amino acids signals in yeast TORC1 pathway. If confirmed, these results would significantly advance our understanding of the TORC1 pathway in yeast and its evolutionary path.

Strength: The authors performed comprehensive IP-MS experiments to probe the interactome of TORC1 in budding yeast, and the resulting datasets are of great interest to researchers in the field of mTOR.

Weakness: The molecular mechanism proposed here is not sufficiently supported by the experiments.

Specifically, the authors propose that there is a direct interaction between Ait1 and TORC1, citing the evidence that they mutually pull down each other in IP-MS experiments. However, additional support for this claim would be needed, such as the identification of residues or mapping of critical domain/motif(s) on Ait1 that physically touch TORC1, native gel or gel-filtration analysis to show the co-migration of Ait1 and TORC1, in vitro binding assays with purified proteins and co-IP experiment to show the interaction between Ait1 and TORC1 is regulated by nitrogen/amino acid availability.

Furthermore, the authors propose that Ait1 and SLC38A9 share sequence homology and therefore they likely bind to Rag/Gtr in a similar way. Additional support for this claim would be needed. For example, it is known that a critical motif in SLC38A9 that mediates the binding with the Rags is an α helix (H60-Y71). In the sequence alignment of Figure 8A, the corresponding region on Ait1 contains no feature of an α helix because multiple serine residues will likely introduce flexibility to prevent α helix formation. The authors would need to provide stronger evidence that this loop directly contacts the Rags. Based on the cryo-EM structure of SLC38A9-Rag, the triple mutant that the authors generated (Ait1c3v1) does not seem appropriate, because none of the three residues directly contact the Rags in the structure, so its defect is difficult to explain.

*Reviewer #3:*

This study uses a proteomics approach to identify proteins that interact directly or indirectly with the Torc1 component Kog1 and microscopy to determine whether these proteins influence the formation of TORC1 bodies, which have been established previously to form on vacuoles in budding yeast upon nitrogen starvation. The authors focus on an interesting new GPCR-like protein identified that they call Ait1, which when deleted results in constitutive formation of TORC1 bodies concomitant with resistance of TORC1 to suppression upon amino acid starvation. Genetic approaches suggest that Ait1 acts upstream of the Gtr1/Gtr2 proteins, known to directly engage TORC1 at the vacuole, for TORC1 suppression and that this suppression might involve sequences within a cytosolic loop of Ait1.

The data are clear that Ait1 represents an important new component of TORC1 regulation by amino acids in *S. cerevisiae* that is not present in most eukaryotic species. The authors make rather specific conclusions regarding the function of Ait1 in the regulation of Gtr1/2 and TORC1. However, its molecular function(s) is incompletely characterized in the study, leaving the conclusions supported by ambiguous data, with several fundamental questions unanswered regarding its placement in the known regulatory network upstream of yeast TORC1.

At the heart of the authors' conclusions on the function of Ait1 is its genetic placement amongst core regulators of TORC1, especially the highly conserved Gtr1/2 proteins. Based on Figure 7, the authors conclude that Ait1 acts upstream of Gtr1/2 for TORC1 regulation (Gtr1/2 are genetically epistatic to Ait1). However, the data in Figure 5 clearly show that the ability of the Gtr1-GTP (GTR1on) and npr2-δ mutants to inhibit TORC1-body formation during nitrogen starvation is dependent on Ait1, thereby placing Ait1 downstream of Gtr1/2 for this regulation.

[Editors' note: further revisions were suggested prior to acceptance, as described below.]

Thank you for resubmitting your work entitled "Ait1 regulates TORC1 signaling and localization in budding yeast" for further consideration by *eLife*. Your revised article has been evaluated by Anna Akhmanova (Senior Editor), the Reviewing Editor, and two expert reviewers.

The manuscript has been improved but there are some remaining issues that need to be addressed, as outlined below:

1) Controls for the co-ip study presented in Figure 3S1 are required, as noted by reviewer 32.

2) The function of Ait1 remains unclear, including the position of this protein in the amino acid sensing pathway, as noted by reviewer #2. The authors should provide a concise summary of their conclusions that is consistent with the data presented that can be used to generate testable hypotheses for future studies.

*Reviewer #1 (Recommendations for the authors):*

The revised manuscript has been improved by removing some of the previously included analysis that was weakly supported by the data presented. The manuscript now describes a better study. However, the biochemical function of Ait1 remains unclear. Nevertheless, the publication of this study will be useful to the field because of the genetic analysis presented.

*Reviewer #2 (Recommendations for the authors):*

In the revised manuscript, Wallace et al. partially addressed some of my previous concerns. However, several new questions arose from this revision.

1. In Figure 3—figure supplement 1, the authors performed a co-IP experiment between Kog and Ait1 in the presence of a crosslinker. First, the authors should show an input panel for Kog and Ait1, as well as a loading control. Second, why does Kog1 show up in three to five bands?

2. The authors claimed that "Ait1 acts at, or above, the level of Gtr1/2 to regulate TORC1", while at the same time, "Ait1 acts at, or below, the level of Gtr1/2 and Pib2 to hold TORC1 in its native position". While some experimental evidence is provided, I am very confused by two claims here. What is the sequence of events upon amino acid supplementation and amino acid deprivation?

---

## [Author Response]

Essential revisions:Revisions required to address the stated conclusions:1) At the heart of the authors' conclusions on the function of Ait1 is its genetic placement amongst core regulators of TORC1, especially the highly conserved Gtr1/2 proteins. Based on Figure 7, the authors conclude that Ait1 acts upstream of Gtr1/2 for TORC1 regulation (Gtr1/2 are genetically epistatic to Ait1). However, the data in Figure 5 clearly show that the ability of the Gtr1-GTP (GTR1on) and npr2-δ mutants to inhibit TORC1-body formation during nitrogen starvation is dependent on Ait1, thereby placing Ait1 downstream of Gtr1/2 for this regulation.

We are sorry that this part of the paper was unclear and believe the confusion stems from the fact that the genetics in Figure 7 shows that Ait1 acts *at or above* the level of Gtr1/2 and the data in Figure 5 shows that Ait1 acts *at or below* the level of Gtr1/2, not simply above Gtr1/2 in Figure 7 and below Gtr1/2 in Figure 5 (we should have used more precise language and have fixed this in the revised manuscript).

To be specific, in Figure 5 we show that deletion of Ait1 leads to the formation of TORC1 foci/bodies, even in nutrient replete conditions. Deletion of Ait1 also completely overrides the severe body formation defects found in npr2, Gtr1on and Pib2 deletion mutants. This places Ait1 as acting at, or below, the level of Gtr1/2 and Pib2 in TORC1 body formation. In contrast, Ait1 does not override the body formation defect seen in a strain carrying mutations in the prion domains of Kog1, indicating that Ait1 acts upstream of the TORC1 agglomeration process itself.

Given that Ait1 interacts with TORC1 (figures 1-3) and Gtr1/2 (new figure 9) it is very likely that Ait1 limits TORC1 body formation by acting *at* the level of TORC1-Gtr1/2. The simplest model is that Ait1 binds to TORC1-Gtr1/2 holding it in place around the vacuolar membrane, and/or by shielding regions of the complex that ultimately drive body formation (such as the prion domains). Once you disrupt the tethering/shielding interaction between Ait1 and TORC1-Gtr1/2, the regulatory events that promote TORC1 body formation during starvation (such as inactivation of Gtr1/2 and TORC1 binding to Pib2) become superfluous.

In Figures 6 and 7 we show that Ait1 is required for TORC1 inactivation in amino acid and leucine starvation in a wild-type strain, but not in strains where Gtr1/2 are locked in the Gtr1-GDP and Gtr2-GTP bound off states or with Gtr1/2 deleted. This places Ait1 as acting at, or above, the level of Gtr1/2 in TORC1 regulation. Again here, given that Ait1 interacts with TORC1-Gtr1/2 and regulates/binds to the complex via a loop that resembles the RagA/C binding domain of SLC38A9, it almost certain that Ait1 regulates Gtr1/2-TORC1 at the level of Gtr1/2.

Thus, all the genetic (and interaction) data fits with a model where Ait1 (i) is bound to TORC1-Gtr1/2 and acts to block its movement into TORC1 bodies until the cell experiences starvation and (ii) helps inactivate TORC1 via Gtr1/2 when cells run short of amino acids. That is, Ait1 acts at the level of TORC1-Gtr1/2 to regulate TORC1 body formation and TORC1 activity.

To clarify this, we have altered the text around Figures 5 and 7 to point out that the data shows that Ait1 acts at or below the level of Gtr1/2 to regulate TORC1 body formation, and at or above the level of Gtr1/2 to regulate TORC1 activity.

2) Page 5-6. The authors suggest that there is a direct interaction between Ait1 and TORC1, citing the evidence that they mutually pull down each other in IP-MS experiments. The following experiments are required to support this conclusion: a) Identify residues or mapping critical domain/motif(s) on Ait1 that required for interaction with TORC1; b) Native gel or gel-filtration analysis to show the co-migration of Ait1 and TORC1; c) in vitro binding assay with purified proteins is preferred and thus recommended; d) Co-IP experiment to show the interaction between Ait1 and TORC1 is regulated by nitrogen/amino acid availability.

We have added Co-IP experiments examining the interaction between Ait1 and TORC1 (Kog1) in detail. These experiments show (i) that the interaction between TORC1 and Ait1 is maintained in starvation conditions (Figure 3s) and (ii) that the interaction between TORC1 and Ait1 can only be detected in extracts treated with cross-linkers (legend Figure 3s)—suggesting the off rate is relatively fast (although this may be caused by the partial degradation of the very large protein Kog1 in extracts seen by us and others). Importantly, however, we have now shown that Ait1 interacts with Gtr1/2 in an extract, even in the absence of a crosslinker (Figure 9 and Figure 9 supplement 1). The Ait1-Gtr1/2 interaction is also constitutive but disrupted by mutations in the C3 loop of Ait1 (Figure 9). More specifically, we see that mutations in the tip of the C3 loop weaken the interaction between Ait1 and Gtr1/2 during log growth conditions, and almost completely block the interaction between Ait1 and Gtr1/2 in amino acid starvation conditions. These data (and signaling data discussed below) strengthen the argument that the interaction between Ait1 and TORC1 (or more accurately the TORC1-Gtr1/2 complex) is close/direct. They also indicate that the interaction between Gtr1/2 and Ait1 involves both the C3 loop and other regions of Ait1.

We also agree that in vitro binding assays are an important next step. However, at this stage we have overwhelming evidence that Ait1 is an important new regulator of Gtr1/2-TORC1 that acts via a very close if not direct interaction with TORC1-Gtr1/2 and have provided clear insight into of Ait1 function. Therefore, given that experiments examining Ait1-Gtr1/2, Ait1-TORC1 and Ait1-TORC1-Gtr1/2 interactions are almost going to be complex since there are multiple proteins and domains involved, the interactions normally occur on a membrane, and Ait1 (and fragments of Ait1) are proving difficult to overexpress and purify, we believe that the work is best left to follow up papers.

3). Figure 8C describes a key experiment suggesting that the ait1C3v1 mutant is a potential separation-of-function mutant, defective in suppressing TORC1 upon leucine starvation but maintaining its ability to prevent TORC1 body formation. However, this effect is never properly examined under nutrient deprivation time courses. Kog1-YFP foci formation over a time course of nitrogen, leucine, and amino acid starvation in this mutant versus wild-type and ait1-δ should be quantified to better demonstrate the effects of this mutant on these two putative functions of Ait1. Effects on pRps6 under nitrogen and amino acid starvation should also be shown.

We have now completely characterized the C3 mutants presented in the paper (Figures 8, 9 and associated supplements). This includes full timecourses of body formation for all C3/C4 mutants (in triplicate), TORC1 activity timecourses in leucine starvation for all C3/C4 mutants, Ait1 localization data for all C3 mutants, and analysis of the impact that the most important C3 mutant (c3v2) has in leucine, complete amino acid, and nitrogen starvation, as well as on the interaction between Gtr1/2 and TORC1.

4) Page 8: The authors suggest that Ait1 and SLC38A9 share sequence homology so they likely bind to Rag/Gtr in a similar way. The following experiments are needed to support this claim: a) A critical motif in SLC38A9 that mediates the binding with the Rags is an α helix (H60-Y71). In the sequence alignment of Figure 8A, the corresponding region on Ait1 contains no feature of an α helix because multiple serine residues will likely introduce flexibility to prevent α helix formation. The authors show provide stronger evidence that this loop directly contacts the Rags; b) Based on the cryo-EM structure of SLC38A9-Rag, the triple mutant that the authors generated (Ait1c3v1) is not appropriate. None of the three residues directly contact the Rags in the structure, so its defect is difficult to explain; c) What is the molecular mechanism of Ait1 function? Does it change the GTP/GDP state of Gtr? Is it a GEF, GDI, or GAP?

These comments were prescient. As described above, the C3v1 mutant presented in the original paper turned out to have a frame shift mutation in its last helix that we missed in the initial round of sequencing. When we remade the mutant (correctly) it had no impact on TORC1 signaling. We therefore went back and looked at the sequences of SLC38A9, Ait1, RagA/C and Gtr1/2, and the RagAC-SLC38A9 structure in more detail. We also examined the conservation of the C3 loop sequence across yeast species. This showed that the residues in the N-terminus of SLC38A9 that align with a highly conserved region of Ait1 around the CIEV sequence (centered at 216-220 in Ait1) make clear contacts with RagC at residues that are conserved in Gtr2 (on a strand that helps form the GTP binding pocket). Other regions are less well conserved. We therefore created three new loop variants, the first with five 5 mutations in the CIEV region, the second with 7 mutations in the CIEV region, and the third introducing nine mutations into a poorly conserved region of the loop. As expected, the first two mutants both had defects in TORC1 regulation (Figure 8). We also showed that these mutations disrupt the interaction between Ait1 and Gtr1/2 (Figure 9).

5) What signal does Ait1 respond to? Does it bind amino acids? Does Ait1 mediate the effects of leucine or other amino acids?

These are very important questions, but we believe best suited for a follow up study. Amino acid metabolism in yeast is highly interconnected and thus strong conclusions about the signals Ait1 responds to have to rely (at least in part) on biochemical studies. It is likely that Ait1 directly binds to leucine, and/or other amino acids, but at least some of this probably occurs via the membrane spanning domains of this GPCR-like protein and thus we have to get this protein correctly overexpressed and folded in a membrane. We will need to identify the binding site (sites), so they are specific since they likely have low affinity for amino acids (which can accumulate up to 1mM concentration in a vacuole) mutate them, and then carry out studies of these mutants in vivo. We are working on this, but due to the complexity of the interactions and system it is going to take a lot of time.

In the possible absence of the requested studies, the conclusions presented should be modified to include only those that are directly supported by experiment.

We have made additional edits to ensure that it is clear that, while we know that Ait1 interacts with TORC1-Gtr1/2 and regulates the complex, we have not proven that this is a direct interaction (and now emphasize binding to TORC1-Gtr1/2 rather than just TORC1 due to the data outlined above). The most important edit is the first sentence of the discussion “In this report, we show that: (i) The previously unstudied GPCR-like protein, Ait1, binds directly or indirectly to TORC1-Gtr1/2 and holds TORC1 around the vacuolar membrane during log-phase growth …” Moreover, in the discussion we start by listing conclusions that are worded to ensure they are directly supported by data. We then present an overall model of Ait1 function that is not fully proven yet (although strongly supported by the data) but then explicitly state that other models can still hold.

[Editors' note: further revisions were suggested prior to acceptance, as described below.]

The manuscript has been improved but there are some remaining issues that need to be addressed, as outlined below:Reviewer #2 (Recommendations for the authors):In the revised manuscript, Wallace et al. partially addressed some of my previous concerns. However, several new questions arose from this revision.1. In Figure 3—figure supplement 1, the authors performed a co-IP experiment between Kog and Ait1 in the presence of a crosslinker. First, the authors should show an input panel for Kog and Ait1, as well as a loading control.

This is a good point, we should have included these controls in the original figure and have now added both a PGK loading control and an Ait1 input panel to Figure 3—figure supplement 1a and b. Note that Ait1 is difficult/impossible to detect in an extract and so the Ait1 input controls are from GFP-Ait1 IPs (carried out in the appropriate cells/conditions) as described in the Methods section. Kog1 is also very difficult to detect in an extract and so the IP material serves as the input control.

Second, why does Kog1 show up in three to five bands?

The 3-5 bands observed are Kog1 (top band) and its degradation products (lower bands). Kog1 is a large (approx. 180kDa) protein with multiple disordered loops and is therefore very susceptible to degradation. We have seen similar patterns in the numerous Kog1-IPs we have carried out over the years and, despite our best efforts, have been unable to block its degradation. For example, as shown in panel c in Figure 3—figure supplement 1, Kog1 is partially degraded even when we increase the concentration of protease inhibitor cocktail to 4x that used in our other experiments. We have made this clear in the figure legend in the line: “Note (i) that Kog1 partially degrades during the IP and thus shows up as multiple bands…”

2. The authors claimed that "Ait1 acts at, or above, the level of Gtr1/2 to regulate TORC1", while at the same time, "Ait1 acts at, or below, the level of Gtr1/2 and Pib2 to hold TORC1 in its native position". While some experimental evidence is provided, I am very confused by two claims here. What is the sequence of events upon amino acid supplementation and amino acid deprivation?

We thank the reviewer for this feedback. Indeed, while we outlined the steps involved in TORC1 inhibition in the discussion we did not integrate the Ait1 dependent regulation of TORC1 localization (TORC1-body formation) into our model. We have therefore edited the section to describe all the steps in TORC1 inhibition/localization. The new section reads:

“Putting our results together with the previously published work showing that the GAP SEAC inactivates Gtr1 during nitrogen and amino acid starvation^28,42-46^, we propose the following model:

First, at the onset of amino acid starvation, SEAC is activated and triggers GTP hydrolysis in Gtr1. The resulting conformational change in Gtr1, and/or signals transmitted through Ait1, then trigger a change in Ait1 structure/activity. Next, Ait1 drives the release of GDP from Gtr2, completing the conversion of Gtr1/2 from the active GTP/GDP-bound state, to the inactive GDP/GTP-bound state. Finally, the resulting conformational change in Gtr1/2, and subsequent movement of the C3 loop in Ait1, exposes a key surface on TORC1 to initiate TORC1-body formation.

As an extension of this model, it may be that strong signals through SEAC, such as those in complete nitrogen starvation (Figure 6—figure supplement 1), override the need for Ait1 due to coupling between the GTP binding sites in Gtr1 and Gtr2^67^.

It is important to point out, however, that while our working model provides a simple explanation for all of the available data (including the observation that deletion/mutation of Ait1 triggers TORC1-body formation), it remains unclear how Ait1 regulates Gtr1/2 and TORC1; Ait1 could also promote TORC1 inhibition by stabilizing the inactive state of Gtr1/2, controlling access to Gtr1/2 activators/repressors (including SEAC), and/or other mechanisms.”

To clarify further, we know (i) that Ait1 binds Gtr1/2, and forms a close interaction with TORC1, in both log growth and starvation conditions and (ii) that deletion or mutation of Ait1 causes TORC1 to move into bodies or aggregates (an event that normally only occurs during starvation) in nutrient rich medium, and blocks TORC1 inhibition during amino acid starvation. Thus, the most parsimonious model holds that once SEAC and Ait1 turn Gtr1/2 off, the resulting conformational change in Gtr1/2 moves the C3 loop in Ait1 to expose a surface of TORC1 that promotes aggregation/body formation. In this model, deletion of Ait1, or mutation of the C3 loop, short circuits the system by constitutively exposing the aggregation domains of TORC1. Following on from the reviewer’s question, this places Ait1 at or below the level of Gtr1/2 (as we see in Figure 5c) since even if you lock Gtr1/2 in their active forms, deleting Ait1 will still remove the C3 loop and expose the TORC1 surfaces that promote aggregation. Our model also places Ait1 as acting at, or above, the level of, Gtr1/2 in the regulation of TORC1 kinase activity (as shown in Figure 7) since locking Gtr1/2 in an off state, or deleting Gtr1 and or 2, bypasses the need for Ait1 in driving TORC1 repression.

It is also important to note that Reviewer 2 specifically asks us to outline the steps that occur in both amino acid supplementation and depletion. However, we only present a model outlining the steps that occur during amino acid depletion since (i) that is what we measure/study in this paper and (ii) the De Virgilio lab has established that TORC1 activation (in amino supplementation) occurs via a distinct mechanism/pathway where the GAP for Gtr2 (Lst4/7) binds the vacuole to initiate reactivation of the Gtr1/2 complex (Pelli-Gulli, 2015 and 2017) and is then driven off the vacuolar membrane once TORC1 is active. In other words, TORC1 inhibition is not simply the reverse of TORC1 activation. In line with this, we recently discovered that the C3 loop of Ait1 is heavily phosphorylated during long term starvation and that this limits Ait1 activity. Thus, further experiments are required to resolve the role of Ait1 in TORC1 reactivation (which appear to be very limited).